# Benchmarking Large Language Models on CMExam - A Comprehensive Chinese Medical Exam Dataset

**Junling Liu**[1][†][*]  **Peilin Zhou**[2][†]  **Yining Hua**[3,4][†]  **Dading Chong**[5]
**Zhongyu Tian**[6]  **Andrew Liu**[5]  **Helin Wang**[7]  **Chenyu You**[8]
**Zhenhua Guo**[9]  **Lei Zhu**[10]  **Michael Lingzhi Li**[4,11]

[1]Alibaba Group [2]Hong Kong University of Science and Technology (Guangzhou)
[3]Harvard University [4]Boston Children's Hospital [5]Peking University
[6]Second Affiliated Hospital of Zhejiang University School of Medicine [7]Johns Hopkins University
[8]Yale University [9]Tianyi Traffic Technology [10]Ant Group [11]Harvard Business School

{william.liuj, zhoupalin, andrew.promed, cszguo, zhulei0305}@gmail.com
1601213984@pku.edu.cn, zhongyutian@zju.edu.cn, hwang258@jhu.edu
yininghua@g.harvard.edu, chenyu.you@yale.edu, mili@hbs.edu

## Abstract

Recent advancements in large language models (LLMs) have transformed the field of question answering (QA). However, evaluating LLMs in the medical field is challenging due to the lack of standardized and comprehensive datasets. To address this gap, we introduce **CMExam**, sourced from the **C**hinese National **M**edical Licensing **Exam**ination. CMExam consists of 60K+ multiple-choice questions for standardized and objective evaluations, as well as solution explanations for model reasoning evaluation in an open-ended manner. For in-depth analyses of LLMs, we invited medical professionals to label five additional question-wise annotations, including *disease groups*, *clinical departments*, *medical disciplines*, *areas of competency*, and *question difficulty levels*. Alongside the dataset, we further conducted thorough experiments with representative LLMs and QA algorithms on CMExam. The results show that GPT-4 had the best accuracy of 61.6% and a weighted F1 score of 0.617. These results highlight a great disparity when compared to human accuracy, which stood at 71.6%. For explanation tasks, while LLMs could generate relevant reasoning and demonstrate improved performance after finetuning, they fall short of a desired standard, indicating ample room for improvement. To the best of our knowledge, CMExam is the first Chinese medical exam dataset to provide comprehensive medical annotations. The experiments and findings of LLM evaluation also provide valuable insights into the challenges and potential solutions in developing Chinese medical QA systems and LLM evaluation pipelines.[1]

## 1 Introduction

Recent advancements brought by large language models (LLMs) such as T5 (Raffel et al., 2020) and GPT-4 (OpenAI, 2023) have revolutionized natural language processing (NLP). However, evaluating LLMs in the medical field poses significant challenges due to the paucity of standardized and comprehensive datasets compiled from reliable and unbiased sources (Li et al., 2023; Zhou et al., 2023b; Hua et al., 2024; Ye et al., 2023; Liu et al., 2023d). Most existing medical datasets (Hendrycks et al., 2020; Abacha et al., 2019b; Li et al., 2023; Zhou et al., 2022) for language model evaluation

---

[*]Corresponding Author. [†]Co-first authors
[1]The dataset and relevant code are available at `https://github.com/williamliujl/CMExam`

37th Conference on Neural Information Processing Systems (NeurIPS 2023) Track on Datasets and Benchmarks.

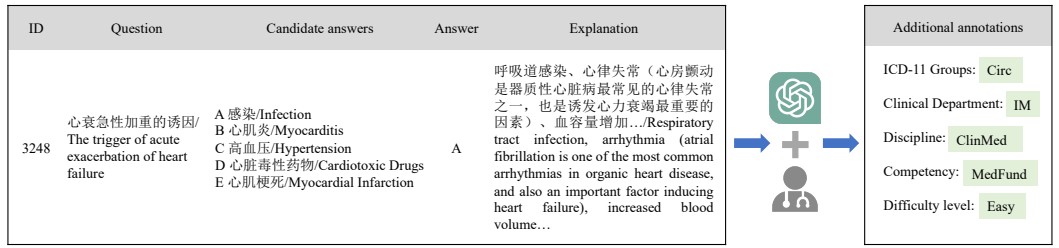

| ID | Question | Candidate answers | Answer | Explanation |
|----|----------|-------------------|--------|-------------|
| 3248 | 心衰急性加重的诱因/ The trigger of acute exacerbation of heart failure | A 感染/Infection
B 心肌炎/Myocarditis
C 高血压/Hypertension
D 心脏毒性药物/Cardiotoxic Drugs
E 心肌梗死/Myocardial Infarction | A | 呼吸道感染、心律失常（心房颤动是器质性心脏病最常见的心律失常之一，也是诱发心力衰竭最重要的因素）、血容量增加…/Respiratory tract infection, arrhythmia (atrial fibrillation is one of the most common arrhythmias in organic heart disease, and also an important factor inducing heart failure), increased blood volume… |

Additional annotations

ICD-11 Groups: Circ
Clinical Department: IM
Discipline: ClinMed
Competency: MedFund
Difficulty level: Easy

Figure 1: An example question of CMExam. Abbreviations: Circulatory System Diseases (Circ), Internal Medicine (IM), Clinical Medicine (ClinMed), Medical Fundamentals (MedFund).

have limitations that hinder comprehensive assessment of LLM performance (Nori et al., 2023). Many datasets are insufficient in terms of size and diversity, preventing a thorough evaluation of LLM capabilities. Furthermore, most datasets primarily focus on text generation tasks rather than utilizing clear choice evaluations, impeding objective and quantitative measurement of LLM performance. Additionally, a majority of these datasets (Li et al., 2023; Pal et al., 2022; Zhu et al., 2020) are sourced from online forums and consumer feedback, which could suffer from significant bias and error. These challenges are particularly amplified in non-English languages, such as Chinese, due to the pervasive inequality in language resources that exists in the NLP field (Bird, 2020; Zeng et al., 2022; Fang et al., 2023). Overall, due to the lack of qualified evaluation datasets, the strengths and weaknesses of LLMs in the medical field have not been fully studied.

In response, we present a novel dataset called CMExam to overcome these challenges and benchmark LLM performance. CMExam is sourced from authentic medical licensing exams. It contains more than 60K questions and utilizes the multiple-choice question format to allow standardized and objective evaluations. Questions in CMExam have corresponding solution explanations that can be used to test LLM's reasoning ability in an open-ended manner. To offer diverse perspectives for measuring LLM performance in the medical field, we created five additional question-wise annotation dimensions based on authenticated resources and objective metrics. To reduce the substantial time and labor costs associated with annotating large-scale datasets, we propose an innovative strategy called GPT-Assisted Annotation. This approach harnessed the power of GPT-4 to automate the initial annotation process. Subsequently, the annotated data underwent a meticulous review and manual verification conducted by two medical professionals. Figure 1 shows an example question from CMExam and the annotation process.

Furthermore, we benchmark the performance of general domain LLMs and medical domain LLMs on answer prediction (multiple-choice) and answer reasoning (open-ended) tasks of CMExam. This comprehensive assessment aims to highlight the strengths and weaknesses of various approaches in Chinese medical QA, with a focus on LLMs. The main findings of this benchmark are as follows:

- GPT-4 (OpenAI, 2023) demonstrates impressive zero-shot performance on the answer prediction task compared to other models, though still significantly lagging behind human performance.
- GPT-3.5 (Brown et al., 2020) and GPT-4 generated reasonable answers on the answer reasoning task despite low BLEU and ROUGE scores. This is because they tended to generate short answers with reasonable quality.
- Existing medical domain LLMs, such as Huatuo (Li et al., 2023) and DoctorGLM (Xiong et al., 2023), exhibit poor zero-shot performance on both tasks, indicating their limited coverage of medical knowledge and substantial room for improvement.
- Lightweight LLMs (e.g., ChatGLM (Du et al., 2022)) fine-tuned on CMExam with supervision chieve performance close to GPT-3.5 on the answer prediction task. They also significantly outperform GPT-3.5 and GPT-4 on the reasoning task while having only 3% of the parameters of GPT-3.5.

In summary, this study provides valuable insights into the performance of LLMs in medical contexts from multiple perspectives, benefiting both the artificial intelligence research community and the medical research community. Our findings contribute to a deeper understanding of the capabilities and limitations of LLMs in the medical domain. Additionally, the CMExam dataset and benchmark introduced in this study serve as valuable resources to inspire researchers to explore more effective

ways of integrating medical knowledge into LLMs, ultimately enhancing their performance in medical applications.

Table 1: A review of medical QA datasets. * indicates availability of additional annotations with authoritative references, † indicates availability of benchmarks, and ‡ indicates datasets with more than 50K questions

| Language | Data Source Type | Question Type | |
|---|---|---|---|
| | | Multiple Choice | Open-ended |
| English | Consumer Questions | MedMCQA (Pal et al., 2022) | LiveQA-Med (Abacha et al., 2017) |
| | | | CliCR‡ (Šuster and Daelemans, 2018) |
| | | | HealthQA (Zhu et al., 2019) |
| | | | MEDIQA (Abacha et al., 2019b) |
| | | | emrQA‡ (Pampari et al., 2018) |
| | | | MedQuaD (Ben Abacha and Demner-Fushman, 2019) |
| | | | MedicationQA* (Abacha et al., 2019a) |
| | | | MEDIQA-AnS (Savery et al., 2020) |
| | | | MASH-QA (Zhu et al., 2020) |
| | Research, Books, or Exams | MEDQA‡(Jin et al., 2021) | BioASQ (Krithara et al., 2023) |
| | | MMLU†‡ (Hendrycks et al., 2020) | MultiMedQA*† (Singhal et al., 2022) |
| | | MedMCQA (Pal et al., 2022) | |
| | | MultiMedQA*† (Singhal et al., 2022) | |
| Chinese | Consumer Questions | - | webMedQA*‡ (He et al., 2019) |
| | | | cMedQA-v1.0‡ (Zhang et al., 2017) |
| | | | cMedQA-v2.0‡ (Zhang et al., 2018) |
| | | | ChiMed (Tian et al., 2019) |
| | | | Huatuo-26M†‡ (Li et al., 2023) |
| | Research, Books, or Exams | MLEC-QA‡ (Zeng et al., 2023a) | MLEC-QA‡ (Zeng et al., 2023a) |
| | | **CMExam**\*†‡(ours) | **CMExam**\*†‡(ours) |

## 2  Related Work

**Medical Question-Answering Datasets**   Table 1 presents a summary of medical QA datasets published after 2017. In particular, we focus on categorizing the data source and question types of the different datasets. Most existing medical QA datasets adopt an open-ended format, primarily because they were constructed directly from consumer questions and answers from doctors. However, multiple-choice and fill-in-the-blank questions provide a more standardized and objective evaluation, and only a small portion of medical QA datasets have adopted these formats. Notable examples include CliCR (Šuster and Daelemans, 2018), MEDQA (Jin et al., 2021), MMLU (Hendrycks et al., 2020), MLEC-QA (Zeng et al., 2023a), and MedMCQA (Pal et al., 2022). Note that the multiple-choice questions in MultiMedQA (Singhal et al., 2022) come from MEDQA, MedMCQA, and MMLU.

Data source types generally determine the reliability of a dataset. Consumer questions collected from web sources require human review to ensure the correctness of the answers. As datasets grow in size, quality control becomes increasingly challenging (Li et al., 2023). In contrast, datasets built from case reports (e.g., CliCR), research literature (e.g., BioAsq (Krithara et al., 2023)), medical books, exams, and related practices (e.g., MMLU and MedMCQA) are often more reliable.

From Table 1, we observe that there are few datasets based on multiple-choice questions from authoritative sources. This characteristic distinguishes CMExam from the MLEC-QA dataset, which is also derived from the Chinese National Medical Licensing Examination. In essence, CMExam has been meticulously crafted as a foundational benchmark dataset. It introduces question explanations for reasoning ability inspection, incorporates expansive annotation facets with authoritative references, and includes question-wise medical competencies and difficulty ratings calculated from human performance. These features make CMExam an indispensable resource for authoritative LLM performance assessment and meaningful human-machine comparisons. Table 2 presents a list of innovations and characteristics of CMExam, which are discussed in detail in the following sections.

**Other Benchmark Datasets of Large Language Models**   The assessment of LLMs has witnessed significant progress, with the introduction of diverse benchmarks that evaluate different dimensions across multiple languages, models and tasks (Liu et al., 2023b,c; Zhou et al., 2023a). Many datasets focus on assessing natural language understanding and reasoning capabilities of LLMs. RACE (Lai et al., 2017) includes English exams for Chinese middle and high school students. TriviaQA (Joshi et al., 2017) consists of question-answer pairs authored by trivia enthusiasts. DROP (Dua et al., 2019)

Table 2: Additional annotations of CMExam.

| Annotation Content | References | Unique values |
|---|---|---|
| Disease Groups | The 11th revision of ICD-11 | 27 |
| Clinical Departments | The Directory of Medical Institution Diagnostic and Therapeutic Categories (DMIDTC) | 36 |
| Medical Disciplines | List of Graduate Education Disciplinary Majors (2022) | 7 |
| Medical Competencies | Medical Professionals | 4 |
| Difficulty Level | Human Performance | 5 |

evaluates reading comprehension with discrete reasoning and arithmetic components. GLUE (Wang et al., 2018) encompasses four existing NLU tasks, while SuperGLUE (Wang et al., 2019) extends it with a more challenging benchmark of eight language understanding tasks. Other datasets, such as HellaSwag (Zellers et al., 2019) and WinoGrande (Sakaguchi et al., 2021), focus on commonsense reasoning. TruthfulQA (Lin et al., 2021) includes health, law, finance, and politics, to assess LLMs' ability to mimic human falsehoods, while MMCU (Zeng, 2023) covers medical, legal, psychology, and education to evaluate multitask Chinese understanding. In addition to language understanding and reasoning, several datasets focus on specific subjects and topics, such as Python coding tasks (Chen et al., 2021), middle school mathematics questions (Cobbe et al., 2021) and defending against attacks (Yi et al., 2023; Xie et al., 2023; Pi et al., 2024). Notably, both C-Eval (Huang et al., 2023) and M3KE (Liu et al., 2023a) serve as multi-level multi-subject evaluation benchmarks, making them particularly suitable for assessing the capabilities of LLMs across multiple domains.

# 3 The CMExam Dataset

**Data Collection and Pre-processing**   CMExam comprises authentic past licensed physician exams in the Chinese National Medical Licensing Examination (CNMLE) collected from the Internet. The CNMLE, also known as the Physician Qualification Examination, is a standardized exam that assesses applicants' medical knowledge and skills in China. It includes a written test with multiple-choice questions covering various medical subjects and a clinical skills assessment simulating patient diagnosis and treatment. We excluded questions that rely on non-textual information, including questions with external information such as images and tables, and questions with keywords "graph" and "table". Duplicate questions were removed from the dataset. In total, 96,161 questions, 68,119 of which were retained after pre-processing. The dataset was then randomly split into training/development/test sets with a ratio of 8:1:1. Each question in the dataset is associated with an ID, five candidate answers, and a correct answer. 85.24% of questions have brief solution explanations and questions in the test set contain additional annotations.

**Data Annotation**   CMExam provides a comprehensive analysis of LLM performance through five additional annotation dimensions. The first dimension involves disease groups based on the 11th revision of the International Classification of Diseases (ICD-11) (World Health Organization (WHO), 2021). ICD-11 is a globally recognized standard classification system for documenting and categorizing health conditions, consisting of 27 major disease groups. The second dimension comprises 36 clinical departments derived from the Directory of Medical Institution Diagnostic and Therapeutic Categories (DMIDTC) [2], published by the National Health Commission of China. DMIDTC is an authoritative guide used for categorizing and naming diagnostic and therapeutic subjects within healthcare institutes. In cases where the question cannot be successfully classified by ICD-11 or DMIDTC, the annotation is marked as "N/A". The third dimension refers to medical disciplines, which are categorized based on the List of Graduate Education Disciplinary Majors (2022) published by the Ministry of Education of the People's Republic of China[3]. This dimension encompasses seven categories representing study majors used in universities. The fourth dimension was created by two medical professionals within the team to assess the primary medical competency tested by each associated question. It consists of four categories. The fifth dimension represents five potential difficulty levels of each question, determined by analyzing the correctness rate observed in human performance data collected alongside the questions. For detailed information on these additional annotations including their potential values, please refer to Table 9, 12, 10, 11. And our proposed GPT-Assisted Annotation strategy is shown in supplementary materials.

---

[2] http://www.nhc.gov.cn/fzs/s3576/201808/345269bd570b47e7aef9a60f5d17db97.shtml
[3] http://www.moe.gov.cn/srcsite/A22/moe_833/202209/t20220914_660828.html

**Dataset Characteristics** The CMExam dataset has several advantages over previous medical QA datasets regarding: 1)*Reliability and Authenticity*: CMExam is sourced exclusively from the CNMLE that undergoes rigorous review and validation processes, ensuring its accuracy and adherence to established medical standards. 2) *Standardization and Comprehensiveness*: CMExam includes both multiple-choice questions that ensure fair and objective evaluations of models' performance and question-wise open-ended reasoning that allows in-depth analysis and assessment of model reasoning abilities and comprehension. Despite the inherent absence of explanations within the CNMLE, we cross-referenced exam questions with solutions offered by diverse online medical examination preparation platforms, effectively enhancing the dataset's informational depth. CMExam reflects the comprehensive coverage of medical knowledge and reasoning required in clinical practice, as it is sourced from carefully designed national medical exams. The inclusion of five additional annotation dimensions enhances the dataset's rigor and offers valuable insights for in-depth evaluation and analysis. 3) *Scale*: CMExam consists of over 60K high-quality questions, providing a large and reliable dataset.

**Data Statistics** The dataset has a total of 68,119 questions, with 65,950 answers being single-choice and 2,169 being multiple-choice, with a maximum of five answer choices. Among all questions, 85.24% have associated solution explanations [3]. Figure 2 shows additional statistics visualization and more basic statistics of CMExam can be seen in supplementary materials. Within the test set, 4,493 questions (65.97%) have corresponding disease group annotations. The most prevalent disease group is Traditional Medicine Disease Patterns (TMDP), followed by Digestive System Diseases, Certain Infectious (Digest) and Parasitic Diseases (InfDis), Endocrine, Nutritional, or Metabolic Diseases (Endo), and Circulatory System Diseases (Circ). For the associated clinical department annotations, 4,965 questions (72.90%) have been assigned values. The two most frequently represented clinical departments are Internal Medicine (IM) and Traditional Chinese Medicine (TCM), with Dentistry (Dent) and Surgery (Surg) following closely. Every question in the test set has been labeled with a discipline, where Clinical Medicine (ClinMed) comprises the largest proportion. Additionally, each question has been categorized into a competency area, with Medical Fundamentals (MedFund) being the predominant category. The difficulty levels of the questions align with common exam patterns, with a greater number of easy questions and a smaller number of hard questions.

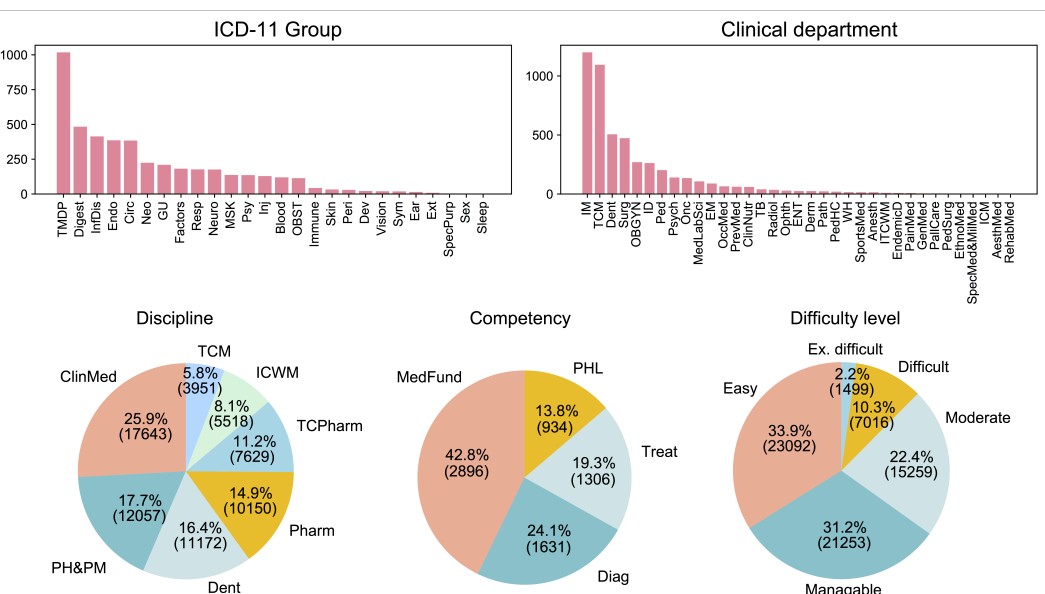

Figure 2: Additional CMExam statistics. For the question length distribution subplot, only the portion within IQR is shown.

---

[3] https://www.yikaobang.com.cn/, http://www.jinyingjie.com/, https://www.lanjiyin.com.cn/

# 4 Benchmarks

## 4.1 Baselines, Settings, and Metrics

**Model Selection**    The LLMs we benchmarked on the CMExam can be divided into two groups based on domains: 1) *General Domain LLMs*: This group comprises GPT3.5/4 (Brown et al., 2020; OpenAI, 2023), ChatGLM (Du et al., 2022; Zeng et al., 2023b), LLaMA (Touvron et al., 2023), Alpaca (Taori et al., 2023), and Vicuna (Chiang et al., 2023). These models are general-purpose language models trained on a massive amount of general-purpose corpora; 2) *Medical Domain LLMs*: This group can be further divided into two subgroups. The first subgroup consists of representative LLMs specifically designed for the medical domain, including DoctorGLM (Xiong et al., 2023) and Huatuo (Wang et al., 2023). DoctorGLM is a healthcare-specific language model initialized with ChatGLM-6B parameters and further fine-tuned on Chinese medical dialogues extracted from ChatGPT. Huatuo, on the other hand, is a knowledge-enhanced model, which builds upon the LLaMA architecture and is additionally supervised-fine-tuned with knowledge-based instruction data harvested from the Chinese medical knowledge graph (CMeKG). The second subgroup comprises medical LLMs that were constructed through supervised fine-tuning of LLMs using the CMExam training set. This subgroup includes models fine-tuned on BERT (Devlin et al., 2019), RoBERTa (Liu et al., 2019), PromptCLUE (Zhang and Xu, 2022) (T5-based), BART  (Shao et al., 2021), Huatuo, ChatGLM, LLaMA, Alpaca, and Vicuna.

**Human Performance**    To effectively gauge the medical proficiency of LLMs, incorporating a measure of human performance into the benchmarking process is of paramount importance. Therefore, during data collection, we preserved the accuracy of human responses for each question. Human performance is estimated by computing a weighted average of response accuracy within each dimension, with weights determined by the number of respondents. This design ensures a robust comparison of LLMs' performance relative to human capabilities, particularly when larger respondent samples contribute to a question's accuracy.

**Experimental Setting**    For GPT models, we leveraged OPENAI's API to access the GPT-3.5-turbo and GPT-4-0314 models, given that their open-source variants are currently unavailable. The LLaMA, Alpaca, and Vicuna models were used in their respective 7B versions, while ChatGLM was evaluated using its publicly accessible 6B version. Additionally, we performed fine-tuning on open-source models using the CMExam dataset. We used P-tuning V2 (Liu et al., 2021) for ChatGLM-6B, with the length of prefix tokens set to 128, and the learning rate set to 2e-2, LoRA (Hu et al., 2021) for LLaMA, Alpaca, Vicuna, and Huatuo models, with the rank set to 8, alpha set to 16, and dropout at 0.05. For BERT models, we followed the fine-tuning methods outlined in (Devlin et al., 2019), with batch size set to 16, learning rate set to 2e-4, hidden dropout probability set to 0.4, and maximum input length set to 192. The fine-tuning processes for all models except BERT involved a batch size of 64, a maximum input length, and a target length of 256. All fine-tuning was performed using NVIDIA V100 GPUs for 10 epochs.

**Metrics**    We assess model performance on multiple choice questions using accuracy and weighted F1 score. These metrics are commonly employed in information retrieval and question-answering tasks to evaluate model performance. For the open-ended solution explanations of CMExam, BLEU (Papineni et al., 2002) and ROUGE (Lin and Hovy, 2003) were used to evaluate the discrepancy between model-generated explanations and ground truth.

## 4.2 Results and Analysis

**Overall Comparison**    We first assessed the performance of general domain LLMs and medical domain LLMs for answer prediction and reasoning tasks. The results are displayed in Table 3. For the answer prediction task, GPT-4 significantly outperforms other methods, demonstrating a zero-shot performance with an accuracy of 61.6% and an F1 score of 0.617. While a performance gap still exists when compared to human performance (which stands at 71.6% accuracy), it's noteworthy that this gap has been greatly reduced from what was observed with GPT-3.5. Among lightweight, general domain LLMs, ChatGLM outperforms LLaMA, Alpaca, and Vicuna, likely attributable to their limited coverage of the Chinese corpus. This restriction seemingly hampers their ability to provide accurate responses to CMExam queries. Furthermore, a noticeable deficiency in zero-shot performance is evident in lightweight medical domain LLMs such as Huatuo, owing to their restricted medical corpus diversity, which hampers the acquisition of broad medical knowledge and

Table 3: Overall comparison on CMExam dataset. We **bold** the best result and underline the second best result.

| Model type | Models | size | Prediction | | Reasoning | | | | |
|---|---|---|---|---|---|---|---|---|---|
| | | | Acc (%) | F1 (%) | BLEU-1 | BLEU-4 | ROUGE-1 | ROUGE-2 | ROUGE-L |
| General Domain | GPT-3.5-turbo | 175B | 46.4±0.6 | 46.1±0.7 | 3.56±0.67 | 1.49±0.51 | 33.80±0.19 | 16.39±0.18 | 14.83±0.13 |
| | GPT-4 | - | **61.6±0.1** | **61.7±0.1** | 0.17±0.00 | 0.06±0.00 | 29.74±0.09 | 14.84±0.04 | 11.51±0.03 |
| | ChatGLM | 6B | 26.3±0.0 | 25.7±0.1 | 16.51±0.08 | 5.00±0.06 | 35.18±0.11 | 15.73±0.05 | 17.09±0.13 |
| | LLaMA | 7B | 0.4±0.0 | 0.3±0.0 | 11.99±0.03 | 5.70±0.0 | 27.33±0.06 | 11.88±0.03 | 10.78±0.04 |
| | Vicuna | 7B | 5.0±0.0 | 4.8±0.1 | 20.15±0.01 | 9.26±0.01 | 38.43±0.02 | 16.90±0.01 | 16.33±0.01 |
| | Alpaca | 7B | 8.5±0.0 | 8.4±0.0 | 4.75±0.00 | 2.50±0.00 | 22.52±0.00 | 9.54±0.00 | 8.40±0.00 |
| Medical Domain | Huatuo | 7B | 12.9±0.0 | 7.0±0.0 | 0.21±0.00 | 0.12±0.00 | 25.11±0.08 | 11.56±0.04 | 9.73±0.02 |
| | MedAlpaca | 7B | 20.0±0.0 | 10.7±0.0 | 0.00±0.00 | 0.00±0.00 | 1.90±0.00 | 0.04±0.00 | 0.52±0.03 |
| | DoctorGLM | 6B | - | - | 9.43±0.09 | 2.65±0.03 | 21.11±0.03 | 6.86±0.01 | 9.99±0.06 |
| | PromptCLUE-base-CMExam | 0.1B | - | - | 18.75±0.08 | 6.65±0.05 | 40.88±0.11 | 21.90±0.11 | 18.31±0.11 |
| | Bart-base-chinese-CMExam | 0.1B | - | - | 23.00±0.40 | 10.35±0.16 | 44.33±0.09 | 24.29±0.09 | 20.80±0.09 |
| | Bart-large-chinese-CMExam | 0.1B | - | - | 26.37±0.18 | 11.65±0.08 | 44.92±0.12 | 24.34±0.12 | 21.75±0.03 |
| | BERT-CMExam | 0.1B | 31.8±0.2 | 31.2±0.2 | - | - | - | - | - |
| | RoBERTa-CMExam | 0.3B | 37.1±0.1 | 36.7±0.4 | - | - | - | - | - |
| | MedAlpaca-CMExam | 7B | 30.5±0.1 | 30.4±0.1 | 16.35±0.80 | 9.78±0.47 | 44.31±0.85 | 27.05±0.50 | 24.55±0.43 |
| | Huatuo-CMExam | 7B | 28.6±0.5 | 29.3±0.2 | 29.04±0.01 | 16.72±0.03 | 43.85±0.24 | 25.36±0.22 | 21.72±0.24 |
| | ChatGLM-CMExam | 6B | 45.3±1.4 | 45.2±1.4 | **31.10±0.23** | **18.94±0.12** | 43.94±0.28 | **31.48±0.14** | **29.39±0.14** |
| | LLaMA-CMExam | 7B | 18.3±0.5 | 20.6±0.5 | 29.25±0.23 | 16.46±0.10 | **45.88±0.04** | 26.57±0.04 | 23.31±0.02 |
| | Alpaca-CMExam | 7B | 21.1±0.6 | 24.9±0.4 | 29.57±0.10 | 16.40±0.12 | 45.48±0.12 | 25.53±0.18 | 22.97±0.06 |
| | Vicuna-CMExam | 7B | 27.3±0.5 | 28.2±0.3 | 29.82±0.03 | 17.30±0.01 | 44.98±0.16 | 26.25±0.13 | 22.44±0.09 |
| Random | Random | - | 3.1±0.2 | 5.1±0.3 | - | - | - | - | - |
| Human Performance | Human volunteers | - | 71.6 | - | - | - | - | - | - |

accurate interpretation of CMExam questions. Our findings suggest that finetuning models with CMExam enhance their performance. For instance, with an accuracy of 45.3%, ChatGLM-CMExam is comparable to GPT-3.5's performance, despite utilizing only about 3% of the parameters employed by GPT-3.5. It is noteworthy that encoder-only LLMs, such as BERT and RoBERTa, remain a robust baseline for answer prediction tasks. Their performance can par with, or even exceed, that of certain decoder-only LLMs, such as LLaMA-CMExam and Alpaca-CMExam, despite having fewer parameters.

For the solution explanation task, we observe that GPT models performed poorly on the BLEU metric, likely due to their tendency to generate short explanations. However, they exhibited an advantage on the ROUGE metric. As DoctorGLM is unable to return answer options according to the prompt, we only report its performance in the solution explanation task. Through finetuning, LLM was able to generate more reasonable explanations. For instance, ChatGLM-CMExam achieved scores of 31.10 and 18.94 on BLEU-1 and BLEU-4, respectively, and scores of 43.94, 31.48, and 29.39 on the ROUGE metrics.

**Results by Disease Groups** Drawing upon ICD-11 annotations (26 categories), we conducted an analysis of the performance of several LLMs across various categories. To mitigate the potential impact of random variability resulting from the number of questions, we limited our analysis to categories containing more than 100 questions. According to Table 4, LLMs have uneven performance and significant gaps in knowledge. GPT-4's accuracy ranges from 74.4% for *Neo* to 44.3% for *TCMDP*, GPT-3.5's accuracy ranges from 63.9% for *Neo* to 31.0% for *TCMDP* and ChatGLM-CMExam's accuracy ranges from 54.7% for *Psy* to 42.9% for *RESP*.

**Results by Clinical Departments** To compare model performance regarding the clinical department dimension (36 categories), we only analyzed categories with more than 50 questions to ensure result representativeness. Results presented in Table 5 highlight that the models show relatively high accuracy on questions associated with commonly encountered departments, such as Emergency Medicine (*EM*), Internal Medicine (*IM*) and Surgery (*Surg*). Their accuracy on questions associated with rarer departments, such as Traditional Chinese Medicine (*TCM*). There is a marked discrepancy in the average accuracy among different departments, with the highest being 50.9% and the lowest being only 13.9%. This observation suggests there are notable variations in medical knowledge and reasoning approaches among different departments. Consequently, it may be necessary to examine specific optimization strategies for different departments.

**Results by Medical Disciplines** Then, we evaluated LLM performance across seven medical disciplines. As depicted in Table 6, the performance of LLMs across disciplines such as Traditional Chinese Medicine (*TCM*), Traditional Chinese Pharmacy (*TCPharm*), and Pharmacy (*Pharm*) was notably subpar, with all accuracy rates falling below 42%. This pattern suggests a potential deficiency

Table 4: Comparing disease classifications.

| Categories | GPT-4 | GPT-3.5 | ChatGLM | ChatGLM-CMExam | Average |
|---|---|---|---|---|---|
| Neo | 74.4±2.2 | 63.9±1.4 | 32.4±1.6 | 51.9±0.2 | 55.6±0.8 |
| Psy | 74.0±0.7 | 62.0±1.7 | 33.3±1.3 | 54.7±0.8 | 56.0±0.9 |
| Factors | 70.0±1.0 | 57.5±1.4 | 28.0±1.1 | 51.1±1.4 | 51.6±0.5 |
| MSK | 65.9±0.8 | 53.8±0.8 | 29.2±0.4 | 53.5±0.0 | 50.6±0.4 |
| GU | 69.2±0.4 | 52.1±1.1 | 30.0±0.2 | 49.5±0.9 | 50.2±0.3 |
| Inj | 65.9±2.3 | 45.7±1.3 | 37.2±2.9 | 49.1±1.8 | 49.5±1.4 |
| Circ | 68.8±0.3 | 49.3±0.7 | 30.9±0.7 | 47.0±0.3 | 49.0±0.2 |
| Endo | 70.6±1.1 | 49.4±1.1 | 25.5±0.8 | 46.1±0.4 | 47.9±0.2 |
| Digest | 67.0±1.0 | 48.8±1.4 | 26.2±0.7 | 49.4±1.1 | 47.8±0.4 |
| InfDis | 66.0±0.5 | 49.2±0.8 | 27.5±0.6 | 48.2±0.8 | 47.7±0.4 |
| Neuro | 64.4±1.2 | 48.7±3.1 | 28.6±0.4 | 45.3±1.3 | 46.7±1.1 |
| OBST | 63.5±0.3 | 45.0±2.4 | 25.7±0.9 | 49.4±0.3 | 45.9±0.5 |
| BLOOD | 69.4±0.3 | 45.3±1.4 | 18.9±1.6 | 43.3±0.7 | 44.2±0.4 |
| Resp | 62.7±0.8 | 44.3±1.4 | 24.5±0.3 | 42.9±0.0 | 43.6±0.7 |
| N/A | 60.0±0.1 | 46.8±0.3 | 24.9±0.2 | 42.5±0.1 | 43.5±0.1 |
| TCMDP | 44.3±0.9 | 31.0±0.6 | 24.2±0.4 | 47.9±0.0 | 36.9±0.6 |

Table 5: Comparing clinical department.

| Categories | GPT-4 | GPT-3.5 | ChatGLM | ChatGLM-CMExam | Average |
|---|---|---|---|---|---|
| EM | 67.4±0.2 | 49.8±0.7 | 36.3±0.4 | 50.2±0.5 | 50.9±0.1 |
| OBGYN | 66.4±1.0 | 51.7±1.5 | 28.6±0.5 | 52.0±0.0 | 49.7±0.3 |
| IM | 70.2±0.6 | 51.8±0.8 | 26.0±1.1 | 47.9±0.9 | 49.0±1.0 |
| ID | 67.4±1.9 | 49.5±3.3 | 26.1±1.9 | 49.6±3.8 | 48.2±1.2 |
| Surg | 63.6±0.8 | 49.5±1.5 | 28.8±0.5 | 47.7±0.9 | 47.4±1.5 |
| ClinNutr | 68.3±2.4 | 48.3±2.9 | 23.9±1.1 | 47.8±0.5 | 47.1±0.7 |
| MedLabSci | 69.2±0.6 | 48.3±2.0 | 29.0±1.5 | 40.8±0.6 | 46.8±0.2 |
| Ped | 64.5±0.0 | 47.2±1.4 | 26.7±2.1 | 41.9±5.5 | 45.1±1.7 |
| N/A | 62.6±0.2 | 48.6±1.1 | 24.6±0.4 | 44.3±0.9 | 45.0±1.0 |
| Ophth | 60.9±0.5 | 39.1±0.8 | 21.8±0.8 | 54.0±0.2 | 44.0±0.8 |
| OccMed | 61.5±4.3 | 38.5±1.6 | 31.3±4.3 | 41.5±3.3 | 43.2±2.5 |
| DENT | 54.9±2.0 | 41.2±1.6 | 27.9±0.8 | 43.5±0.9 | 41.9±1.0 |
| TCM | 43.1±1.3 | 31.4±1.3 | 24.5±1.9 | 45.8±4.4 | 36.2±0.6 |
| ENT | 41.3±0.8 | 28.0±0.6 | 29.3±0.1 | 26.7±0.1 | 31.3±0.5 |
| ICM | 33.3±0.0 | 11.1±15.7 | 0.0±0.0 | 11.1±15.7 | 13.9±4.8 |

in the exposure of these models to data within these categories. Conversely, disciplines such as *ClinMed* and *Ph&PM* demonstrated higher accuracy rates, likely due to the abundance of relevant data. The observed variability in performance across different disciplines underscores the distinctiveness of data characteristics and complexities inherent to each field, thereby advocating for discipline-specific model optimizations and enhancements.

**Results by Competencies** Evaluations based on medical competency areas aimed at a higher-level understanding of model capability in solving medical problems. As indicated in Table 7, the lowest average accuracy across LLMs was observed within the domain of mastering Medical Fundamentals (*MedFund*), with a meager average score of 42.1%. This result demonstrates that these models, predominantly trained on general textual data, have inadequate exposure to medical-specific data. While fine-tuning did provide some improvement, these models could benefit from additional medical scenario data to further augment their performance. It is worth highlighting that the average accuracy in the domain of Public Health Laws and Ethics (*PHL*) was reasonably high, notably achieving an average of 47.6%. In addition, the LLMs showcased their proficiency in accurate disease diagnosis.

**Results by Question Difficulty** To evaluate model performance in tackling questions of varying levels of difficulty, we conducted experiments regarding the question difficulty dimension, which was calculated based on human exam-taker performance. As shown in Table 8, there's an evident trend where model accuracies decrease as question complexity rises. This pattern suggests that more sophisticated questions demand an extensive knowledge base and complex reasoning, which are challenging for the LLMs, thus reflecting patterns observed in human performance.

Table 6: Comparing medical discipline.

| Categories | GPT-4 | GPT-3.5 | ChatGLM | ChatGLM-CMExam | Average |
|---|---|---|---|---|---|
| ClinMed | 67.9±0.1 | 51.4±0.4 | 27.3±0.3 | 48.9±0.4 | 48.8±0.7 |
| PH&PM | 68.2±0.4 | 52.7±1.7 | 26.2±0.3 | 47.3±1.0 | 48.6±0.5 |
| ICWM | 56.1±0.1 | 40.0±2.3 | 29.4±0.8 | 53.6±0.7 | 44.8±0.9 |
| Dent | 59.5±0.7 | 43.9±1.9 | 28.5±1.1 | 45.3±0.6 | 44.3±0.3 |
| Pharm | 61.1±0.4 | 46.3±0.5 | 23.2±0.2 | 37.0±0.1 | 41.9±0.3 |
| TCM | 53.5±0.4 | 35.9±0.2 | 24.1±0.3 | 49.1±0.0 | 40.6±1.1 |
| TCPharm | 45.4±1.2 | 35.6±0.1 | 24.1±1.0 | 43.1±0.4 | 37.1±0.5 |

Table 7: Comparing LLMs' competencies.

| Categories | GPT-4 | GPT-3.5 | ChatGLM | ChatGLM-CMExam | Average |
|---|---|---|---|---|---|
| Diag | 70.1±5.5 | 50.9±2.1 | 30.9±2.8 | 51.6±1.0 | 50.9±1.4 |
| PHL | 64.2±0.7 | 50.0±0.5 | 26.8±0.3 | 49.6±0.1 | 47.6±0.3 |
| Treat | 56.5±0.5 | 43.0±1.1 | 25.7±0.2 | 47.4±0.6 | 43.2±0.8 |
| MeFund | 58.3±0.3 | 44.6±0.7 | 23.9±0.5 | 41.6±0.4 | 42.1±0.9 |
| N/A | 54.8±0.2 | 30.4±0.4 | 23.7±0.1 | 38.5±0.2 | 36.9±0.3 |

Table 8: Results by question difficulty.

| Categories | GPT-4 | GPT-3.5 | ChatGLM | ChatGLM-CMExam | Average |
|---|---|---|---|---|---|
| Easy | 74.6±0.1 | 58.5±0.6 | 31.4±0.2 | 61.5±0.3 | 56.5±0.4 |
| Manageable | 63.9±0.2 | 47.4±0.7 | 25.9±0.5 | 46.1±0.3 | 45.8±0.6 |
| Moderate | 51.3±0.6 | 36.8±0.8 | 23.0±0.4 | 34.5±0.6 | 36.4±0.7 |
| Difficult | 36.4±0.9 | 26.2±0.7 | 18.9±0.5 | 24.3±0.9 | 26.5±0.6 |
| Extremely difficult | 27.2±1.0 | 21.4±2.2 | 15.8±1.0 | 12.2±1.1 | 19.1±1.1 |

**Results by Question Length**    Finally, to investigate if model performance is associated with input lengths, we compared their performance regarding question lengths. Figure 3 illustrates that Large Language Models (LLMs) generally show higher accuracy with problem lengths between 60 and 90. However, their performance seems to falter with problems that are either too short or overly long. Additionally, we noticed that the effect of question length varies across different LLMs. For instance, GPT models tend to incrementally improve as the problem length expands, performing optimally within the 50 to 90 range. Conversely, ChatGLM-CMExam's performance fluctuates noticeably with varying lengths, and it tends to fall short compared to GPT models when addressing longer problems.

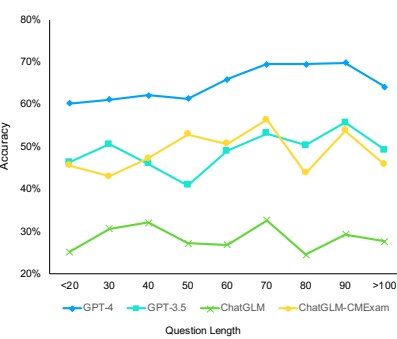

Figure 3: Results stratified by question length.

## 5    Conclusion and Discussions

In this work, we developed CMExam, a dataset sourced from the stringent Chinese National Medical Licensing Examination, featuring 60,000+ multiple-choice questions, with detailed explanations. CMExam ensures reliability, validity, and adherence to medical standards. It also demonstrates the practicality of employing GPT-4 to automate the annotation process, which strikes a harmonious balance between efficiency and cost-effectiveness while maintaining the desired level of accuracy and reliability of the annotation. Utilizing this large and reliable corpus, we tested several LLMs for answer selection and reasoning tasks. A performance gap was observed between LLMs and human experts, signaling the need for additional LLM research. CMExam's standardization and comprehensiveness also ensure objective evaluations of models while enabling in-depth analysis of their reasoning capabilities. The questions cover a wide spectrum of medical knowledge, augmented with five additional annotation dimensions for rigorous evaluation. This study aims to spur further exploration of LLMs in medicine by providing a comprehensive benchmark for their evaluation.

We anticipate CMExam to contribute significantly to future advancements of LLMs, particularly in handling medical question-answering tasks.

**Limitations** Firstly, while CMExam is derived from meticulously designed medical examinations, our process of excluding questions requiring non-textual information may inadvertently affect the balance of the remaining questions, potentially introducing unexpected biases. It is critical to acknowledge this aspect while interpreting any findings or analyses conducted using this dataset. Furthermore, the current BLEU and ROUGE metrics primarily evaluate the explanation task, but these measures are insufficient for assessing the reasonableness of the answer. In future work, we will incorporate human evaluation to provide a more comprehensive assessment of the models.

**Ethics** CMExam is a dataset derived from the Chinese National Medical Licensing Examination, which aligns with numerous datasets containing similar National Medical Licensing Examinations (Zeng et al., 2023a; Hendrycks et al., 2020; Jin et al., 2021; Pal et al., 2022; Singhal et al., 2022). We have ensured adherence to applicable legal and ethical guidelines during data collection and use. The authenticity and accuracy of the exam questions have been thoroughly verified, providing a reliable basis for evaluating LLMs. Please note that the CMExam dataset is intended for academic and research purposes only. Any commercial use or other misuse that deviates from this purpose is expressly prohibited. We urge all users to respect this stipulation in the interest of maintaining the integrity and ethical use of this valuable resource.

**Societal Impacts** While CMExam aims to enhance LLM evaluations in the medical field, it should not be misused for assessing individual medical competence or for patient diagnosis. Conclusions drawn from models trained on this dataset should acknowledge its limitations, especially given its single source and the specific context of the CNMLE. The use of this dataset should strictly be limited to research purposes to avoid potential misuse.

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

# A  Appendix

## A.1  Abbreviations, Full Names, and Translations of Additional Annotations

This section presents four tables of additional annotations that contain translation. It showcases abbreviations, full English names, and Chinese names for each group in each annotation dimension. Table 9 showcases all disease groups included in the 11th revision of the International Classification of Diseases (ICD-11). We present the disease group in the same order found on the official website. Table 12 offers a classification of 36 clinical departments derived from the Directory of Medical Institution Diagnostic and Therapeutic Categories. Table 10 presents a breakdown of medical disciplines based on the List of Graduate Education Disciplinary Majors published by the Ministry of Education of the People's Republic of China. This categorization comprises seven study majors used in universities. Table 11 provides all groups of areas of medical competency assessed in Chinese medical licensing exams.

Table 9: ICD-11 Groups

| Code | Abbreviation | Full English Name | Chinese Name |
|------|-------------|-------------------|--------------|
| 01 | InfDis | Certain infectious or parasitic diseases | 某些感染性疾病或寄生虫病 |
| 02 | Neo | Neoplasms | 肿瘤 |
| 03 | Blood | Diseases of the blood or blood-forming organs | 血液或造血器官疾病 |
| 04 | Immune | Diseases of the immune system | 免疫系统疾病 |
| 05 | Endo | Endocrine, nutritional or metabolic diseases | 内分泌、营养或代谢疾病 |
| 06 | Psy | Mental, behavioural or neurodevelopmental disorders | 精神、行为或神经发育障碍 |
| 07 | Sleep | Sleep-wake disorders | 睡眠-觉醒障碍 |
| 08 | Neuro | Diseases of the nervous system | 神经系统疾病 |
| 09 | Vision | Diseases of the visual system | 视觉系统疾病 |
| 10 | Ear | Diseases of the ear or mastoid process | 耳或乳突疾病 |
| 11 | Circ | Diseases of the circulatory system | 循环系统疾病 |
| 12 | Resp | Diseases of the respiratory system | 呼吸系统疾病 |
| 13 | Digest | Diseases of the digestive system | 消化系统疾病 |
| 14 | Skin | Diseases of the skin | 皮肤疾病 |
| 15 | MSK | Diseases of the musculoskeletal system or connective tissue | 肌肉骨骼系统或结缔组织疾病 |
| 16 | GU | Diseases of the genitourinary system | 泌尿生殖系统疾病 |
| 17 | Sex | Conditions related to sexual health | 性健康相关情况 |
| 18 | OBST | Pregnancy, childbirth or the puerperium | 妊娠、分娩或产褥期 |
| 19 | Peri | Certain conditions originating in the perinatal period | 起源于围生期的某些情况 |
| 20 | Dev | Developmental anomalies | 发育异常 |
| 21 | Sym | Symptoms, signs or clinical findings, not elsewhere classified | 症状、体征或临床所见，不可归类在他处者 |
| 22 | Inj | Injury, poisoning or certain other consequences of external causes | 损伤、中毒或外因的某些其他后果 |
| 23 | Ext | External causes of morbidity or mortality | 疾病或死亡的外因 |
| 24 | Factors | Factors influencing health status or contact with health services | 影响健康状态或与 |
| 25 | SpecPurp | Codes for special purposes | 用于特殊目的的编码 |
| 26 | TCMDP | Supplementary Chapter Traditional Medicine Conditions - Module I | 补充章传统医学病证-模块1 |
| V | FuncAssess | Supplementary section for functioning assessment | 功能评定补充部分 |
| X | ExtCodes | Extension Codes | 扩展码 |
| - | N/A | Not Applicable | 不符合 |

Table 10: Medical Disciplines

| Abbreviation | Full English Name | Chinese Name |
|------|-------------------|--------------|
| ClinMed | Clinical Medicine | 临床医学 |
| Dent | Dentistry | 口腔医学 |
| ICWM | Integrated Chinese and Western Medicine | 中西医结合 |
| PH&PM | Public Health and Preventive Medicine | 公卫预防 |
| Pharm | Pharmacy | 药学 |
| TCM | Traditional Chinese Medicine | 中医学 |
| TCPharm | Traditional Chinese Pharmacy | 中药学 |

Table 11: Areas of competencies

| Abbreviation | Full English Name | Chinese Name |
|------|-------------------|--------------|
| Diag | Disease Diagnosis and Differential Diagnosis | 疾病诊断和鉴别诊断 |
| MedFund | Medical Fundamentals | 医学基础知识 |
| N/A | Not Applicable | 不符合 |
| PHL | Public Health Law and Ethics | 公共卫生法律伦理 |
| Treat | Disease Treatment | 疾病治疗 |

Table 12: Clinical Departments

| Abbreviation | Full English Name | Chinese Name |
|---|---|---|
| AesthMed | Aesthetic Medicine | 医疗美容科 |
| Anesth | Anesthesiology | 麻醉科 |
| ClinNutr | Clinical Nutrition | 临床营养科 |
| Dent | Dentistry | 口腔科 |
| Derm | Dermatology | 皮肤科 |
| EM | Emergency Medicine | 急诊医学科 |
| EndemicD | Endemic Disease | 地方病科 |
| ENT | Otolaryngology | 耳鼻咽喉科 |
| EthnoMed | Ethnic Medicine | 民族医学科 |
| GenMed | General Medicine | 全科医疗 |
| ICM | Intensive Care Medicine | 重症医学科 |
| ID | Infectious Diseases | 传染科 |
| IM | Internal Medicine | 内科 |
| ITCWM | Integrated Traditional Chinese and Western Medicine | 中西医结合科 |
| MedLabSci | Medical Laboratory Science | 医学检验科 |
| N/A | Not Applicable | 不符合 |
| OBGYN | Obstetrics and Gynecology | 妇产科 |
| OccMed | Occupational Medicine | 职业病科 |
| Onc | Oncology | 肿瘤科 |
| Ophth | Ophthalmology | 眼科 |
| PainMed | Pain Medicine | 疼痛科 |
| PallCare | Palliative Care | 临终关怀科 |
| Path | Pathology | 病理科 |
| Ped | Pediatrics | 儿科 |
| PedHC | Pediatric Health Care | 儿童保健科 |
| PedSurg | Pediatric Surgery | 儿童外科 |
| PrevMed | Preventive Medicine | 预防保健科 |
| Psych | Psychiatry | 精神科 |
| PT | Physical Therapy | 理疗科 |
| Radiol | Radiology | 医学影像科 |
| RehabMed | Rehabilitation Medicine | 康复医学科 |
| SpecMed&MilMed | Special Medical and Military Medicine | 特种医学与军事医学科 |
| SportsMed | Sports Medicine | 运动医学科 |
| Surg | Surgery | 外科 |
| TB | Tuberculosis | 结核病科 |
| TCM | Traditional Chinese Medicine | 中医科 |
| WH | Women's Health | 妇女保健 |

## A.2 Instructions for Pre-annotation

In this section, we present instructions used to pre-annotate CMExam test set data using GPT4. As shown in Figure 4,5,6,7, we first constrained the output from GPT4 to return only specific categories. We then annotated each of the five additional annotation dimensions relevant to this study with all the category information for each dimension. Next, we provided specific prompt information and finally, we performed filtering on the GPT4 output to improve the effectiveness of pre-annotation. During the actual annotation process, specific categories and prompt information should be filled in the grey background areas.

ZH:返回格式限制为某个具体类目的名称即可。
EN:The return format is limited to the name of a specific category.

ZH:共有27个类别：
某些传染病或寄生虫病; 肿瘤; 血液或造血器官的疾病; 免疫系统疾病; 内分泌、营养或代谢疾病,精神、行为或神经发育障碍; 睡眠-清醒障碍; 神经系统疾病; 视觉系统疾病; 耳或乳突疾病; 循环系统疾病; 呼吸系统疾病; 消化系统疾病; 皮肤疾病; 肌肉骨骼系统或结缔组织疾病; 泌尿生殖系统疾病; 与性健康有关的情况; 妊娠、分娩或产褥期;围生期某些疾患;发育异常;其他未分类的症状、体征或临床表现;损伤、中毒或外部原因引起的其他后果;影响健康状况的因素或与卫生服务的接触; 传统医学疾病;功能评估补充部分;扩展代码;疾病或死亡的不确定或未知原因.
EN: There are twenty-seven categories: Certain infectious or parasitic diseases; Neoplasms; Diseases of the blood or blood-forming organs; Diseases of the immune system; Endocrine, nutritional or metabolic diseases; Mental, behavioral or neurodevelopmental disorders; Sleep-wake disorders; Diseases of the nervous system; Diseases of the visual system; Diseases of the ear or mastoid process; Diseases of the circulatory system; Diseases of the respiratory system; Diseases of the digestive system; Diseases of the skin; Diseases of the musculoskeletal system or connective tissue; Diseases of the genitourinary system; Conditions related to sexual health; Pregnancy, childbirth or the puerperium; Certain conditions originating in the perinatal period; Developmental anomalies; Symptoms, signs or clinical findings, not elsewhere classified; Injury, poisoning or certain other consequences of external causes; Factors influencing health status or contact with health services; Traditional Medicine conditions; Supplementary section for functioning assessment; Extension codes; Uncertain or unknown cause of morbidity or mortality.

ZH: 假设你是一位医疗行业专家，请判断下面这个题目属于哪个ICD-11的类别，若都不符合，则只返回"不符合"这个标签。
EN: Assuming you are an expert in the medical industry, please determine which ICD-11 category this question belongs to. If none of the categories apply, return the label "N/A"

ZH:题目信息为"女34岁。月经量进行性减少，现闭经半年，泌乳3个月，首选检查项目应是：A 孕激素试验，B 血HCG测定，C 血PRL测定，D 性激素测定，E 诊断性刮宫"。
EN:The question is "A 34-year-old woman has experienced progressive reduction in menstrual flow and has been amenorrheic for 6 months. She has been lactating for 3 months. Which of the following is the preferred test to perform? A. Progesterone test B. Blood HCG test C. Blood PRL test D. Sex hormone test E. Diagnostic curettage".

ZH:注意，不需要回答问题本身，只需要返回这个题目与上述27个类目中的哪个类目最相关，返回27个类目中的一个，不需要其他文字。
EN:Note that you do not need to answer the question itself, just return which of the twenty-seven categories listed above is most relevant to this question. Return only one of the twenty-seven categories, no additional words necessary.

Figure 4: Pre-annotation Instructions for Disease Groups.

ZH:返回格式限制为某个具体类目的名称即可。
EN:The return format is limited to the name of a specific category.

ZH:共有36个类别：
预防保健科; 全科医疗; 内科; 外科; 妇产科; 妇女保健; 儿科; 儿童外科; 儿童保健科; 眼科; 耳鼻咽喉科; 口腔科; 皮肤科; 精神科; 传染科; 肿瘤科; 急诊医学科; 中医科; 结核病科; 疼痛科; 医疗美容科; 地方病科; 康复医学科; 理疗科; 运动医学科; 职业病科; 特种医学与军事医学科; 临终关怀科; 临床营养科; 中西医结合科; 民族医学科; 麻醉科; 医学检验科; 病理科; 医学影像科; 重症医学科.
EN: There are thirty-six categories: Preventive Medicine; General Medicine; Internal Medicine; Surgery; Obstetrics and Gynecology; Women's Health; Pediatrics; Pediatric Surgery; Pediatric Health Care; Ophthalmology; Otolaryngology; Dentistry; Dermatology; Psychiatry; Infectious Diseases; Oncology; Emergency Medicine; Traditional Chinese Medicine; Tuberculosis; Pain Medicine; Aesthetic Medicine; Endemic Disease; Rehabilitation Medicine; Physical Therapy; Sports Medicine; Occupational Medicine; Special Medical and Military Medicine; Palliative Care; Clinical Nutrition; Integrated Traditional Chinese and Western Medicine; Ethnic Medicine; Anesthesiology; Medical Laboratory Science; Pathology; Radiology; Intensive Care Medicine.

ZH: 假设你是一位医疗行业专家，请判断下面这个题目属于哪个DMIDTC的类别，若都不符合，则只返回"不符合"这个标签。
EN: Assuming you are an expert in the medical industry, please determine which DMIDTC category this question belongs to. If none of the categories apply, return the label "N/A"

ZH:题目信息为"女34岁。月经量进行性减少，现闭经半年，泌乳3个月，首选检查项目应是：A 孕激素试验，B 血HCG测定，C 血PRL测定，D 性激素测定，E 诊断性刮宫"。
EN:The question is "A 34-year-old woman has experienced progressive reduction in menstrual flow and has been amenorrheic for 6 months. She has been lactating for 3 months. Which of the following is the preferred test to perform? A. Progesterone test B. Blood HCG test C. Blood PRL test D. Sex hormone test E. Diagnostic curettage".

ZH:注意，不需要回答问题本身，只需要返回这个题目与上述36个类目中的哪个类目最相关，返回36个类目中的一个，不需要其他文字。
EN:Note that you do not need to answer the question itself, just return which of the thirty-six categories listed above is most relevant to this question. Return only one of the thirty-six categories, no additional words necessary.

Figure 5: Pre-annotation Instructions for Clinical Departments.

Figure 6: Pre-annotation Instructions for Medical Disciplines.

ZH:共有4个类别：医学基础知识、疾病诊断和鉴别诊断、疾病治疗、公共卫生法律伦理。
EN:There are four categories: Basic Medical Knowledge, Disease Diagnosis and Differential Diagnosis, Disease Treatment, and Public Health Law and Ethics.

Figure 7: Pre-annotation Instructions for Areas of Competencies.

## A.3 Analysis of Model Generation Ability

In Figure 8, we present partial explanations generated by various models for a medical question from the CMExam dataset. Notably, GPT-4 and GPT-3.5 produce concise and sensible explanations, which may account for the lower BLUE scores. Conversely, models like Vicuna, LLaMA, and Huotuo demonstrate a more prominent repetition phenomenon, while Alpaca simply duplicates the provided options without providing an explanation.

Fine-tuning models on the CMExam dataset significantly reduces the repetition phenomenon and improves the overall reasonableness of the explanations. For instance, the ChatGLM-CMExam model analyzes each option in a similar manner to the solution explanation. However, some models still generate unreasonable explanations, as observed in LLaMA-CMExam, Alpaca-CMExam, and Vicuna-CMExam. This could be attributed to their training on generic data and lack of specific knowledge in the medical domain. This underscores the significance of training large language models with a focus on the medical domain.

| Chinese | Translated English |
|---|---|
| **Question** | |

<table>
<tr><td colspan="3">Question</td></tr>
<tr><td>Question</td><td>除有关节肿痛外，对类风湿关节炎诊断最有意义的表现
A 足跟足掌部位痛
B 关节隆起部与受压部位皮下出现无痛性结节
C 弥漫性肺间质病变
D 胸腔积液（糖含量正常）
E 小腿痛性皮下结节</td><td>In addition to joint swelling and pain, the most significant manifestations for the diagnosis of rheumatoid arthritis is:
A. Pain in the heel and sole of the foot
B. Painless nodules under the skin at the sites of joint prominence and pressure
C. Diffuse pulmonary interstitial disease
D. Pleural effusion (with normal glucose content)
E. Painful subcutaneous nodules in the lower leg</td></tr>
<tr><td colspan="3">Ground Truth</td></tr>
<tr><td>Solution Explanation</td><td>类风湿关节炎最常出现的部位为腕、掌指、近端指间关节，足跟足掌部位痛（A错）一般不见于类风湿关节炎，故非最有意义的临床表现。关节外的表现为类风湿结节，关节隆起与受压部位常见，为无痛性皮下结节（B对），其典型的病理改变可有助于类风湿关节炎的诊断。肺受累时常表现为肺间质改变，肺内可出现单个或多个结节，但非最有意义的临床表现（C错）。类风湿关节炎出现胸膜炎（P808）时表现为单侧或双侧性的少量胸腔积液，胸水呈渗出性，糖含量很低（D错）。小腿痛性皮下结节（E错）在类风湿关节炎中不会出现。</td><td>The most common sites of rheumatoid arthritis are the wrists, fingers, and proximal interphalangeal joints. Pain in the heel and sole of the foot (A false) is generally not seen in rheumatoid arthritis, so it is not the most significant clinical manifestation. Extra-articular manifestations include rheumatoid nodules, which are commonly found at joint prominences and areas under pressure. They are painless subcutaneous nodules (B true), and their typical pathological changes can help with the diagnosis of rheumatoid arthritis. Pulmonary involvement is often characterized by interstitial changes in the lungs. Single or multiple nodules may appear in the lungs, but they are not the most significant clinical manifestation (C false). When pleurisy (P808) occurs in rheumatoid arthritis, it presents as a small amount of pleural effusion on one or both sides of the chest. The pleural fluid is exudative with a very low glucose content (D false). Painful subcutaneous nodules in the lower legs (E false) do not occur in rheumatoid arthritis.</td></tr>
<tr><td colspan="3">Model Generation</td></tr>
<tr><td>GPT-4</td><td>关节隆起部与受压部位皮下出现无痛性结节是类风湿关节炎诊断的一个重要表现。</td><td>Painless nodules under the skin at the sites of joint prominence and pressure are an important manifestation of the diagnosis of rheumatoid arthritis.</td></tr>
<tr><td>GPT-3.5</td><td>类风湿关节炎最有意义的表现是关节隆起部与受压部位皮下出现无痛性结节。这些结节通常出现在关节附近大小为米粒大小至拳头大小。这些结节被称作类风湿瘤与类风湿关节炎的严重程度和预后相关。</td><td>The most significant manifestation of rheumatoid arthritis is the presence of painless nodules under the skin at the sites of joint prominence and pressure. These nodules typically appear near the joints and range in size from rice grain to fist size. These nodules are called rheumatoid nodules and are associated with the severity and prognosis of rheumatoid arthritis.</td></tr>
<tr><td>ChatGLM</td><td>对于类风湿关节炎的诊断，最有价值的表现是关节隆起部与受压部位皮下出现无痛性结节。这种结节通常是在关节周围的皮肤上出现的，可能是由关节周围的炎症细胞产生的。其他选项，足跟足部位痛和弥漫性肺间质病变可能是类风湿关节炎的其他常见表现但是对诊断类风湿关节炎的价值不如关节隆起部与受压部位皮下出现无痛性结节。胸腔积液(糖含量正常)和小腿痛性皮下结节虽然也是类风湿关节炎的表现之一，但它们通常不是最有价值的表现。</td><td>The most valuable manifestation for the diagnosis of rheumatoid arthritis is the presence of painless nodules under the skin at the sites of joint prominence and pressure. These nodules typically appear on the skin around the joints and may be produced by inflammatory cells surrounding the joints. While pain in the heel and sole of the foot and diffuse interstitial lung disease are other common manifestations of rheumatoid arthritis, they are not as valuable for diagnosing rheumatoid arthritis as the presence of painless nodules in the swollen and pressured areas of the joints. Pleural effusion (with normal glucose levels) and painful subcutaneous nodules in the legs are also manifestations of rheumatoid arthritis, but they are typically not the most valuable manifestations.</td></tr>
<tr><td>Vicuna</td><td>A 足跟足掌部位痛
B 关节隆起部与受压部位皮下出现无痛性结节
C 弥漫性肺间质病变
D 胸腔积液（糖含量正常）
E 小腿痛性皮下结节，
答案是A 足跟足掌部位痛。
这个问题的答案是A 足跟足掌部位痛。
这个问题的答案是A 足跟足掌部位痛。</td><td>A. Pain in the heel and sole of the foot.
B. Painless nodules under the skin at the sites of joint prominence and pressure.
C. Diffuse pulmonary interstitial disease.
D. Pleural effusion (with normal glucose content).
E. Painful subcutaneous nodules in the lower leg,
The answer is A, pain in the heel and sole of the foot.
The answer to this question is A, pain in the heel and sole of the foot.
The answer to this question is A, pain in the heel and sole of the foot.</td></tr>
<tr><td>LLaMA</td><td>1. A 2. B 3. C 4. D 5. E 6. A 7. B</td><td>1. A 2. B 3. C 4. D 5. E 6. A 7. B</td></tr>
<tr><td>Alpaca</td><td>A 足跟足掌部位痛
B 关节隆起部与受压部位皮下出现无痛性结节
C 弥漫性肺间质病变
D 胸腔积液（糖含量正常）
E 小腿痛性皮下结节</td><td>A. Pain in the heel and sole of the foot
B. Painless nodules under the skin at the sites of joint prominence and pressure
C. Diffuse pulmonary interstitial disease
D. Pleural effusion (with normal glucose content)
E. Painful subcutaneous nodules in the lower leg</td></tr>
<tr><td>Huotuo</td><td>肿瘤、疼痛、弥漫、结节、积液、糖尿病、感染、疼痛、结节、疼痛、结节、疼痛、结节、结节、结节、结节、结节、结节、结节、结节、结节、结节、结节、结节、结节、结节、结节、结节、结节、结节、结节、结节、结节、结节</td><td>Tumor, pain, diffuse, nodules, effusion, diabetes, infection, pain, nodules, pain, nodules, pain, nodules, nodules, nodules, nodules, nodules, nodules, nodules, nodules, nodules, nodules, nodules, nodules, nodules, nodules, nodules, nodules, nodules, nodules, nodules</td></tr>
<tr><td>Huotuo-CMExam</td><td>类风湿关节炎的诊断重点是无痛性结节，如有痛则可诊断为类湿或类疾病，如有痛的类风湿关节炎，可诊断为类风湿关节炎。掌握"类风湿关节炎"知识点。<unk>inaquirepod/i/11199944982222222222222222222222222222222222222222222222222222222222222222222</td><td>The key for diagnosing rheumatoid arthritis is the presence of painless nodules. If there is pain, it may be diagnosed as rheumatism or a similar disease. If there is rheumatoid arthritis with pain, it may be diagnosed as rheumatoid arthritis. Have a good understanding of the knowledge related to "rheumatoid arthritis."
<unk>inaquirepod/i/11199944982222222222222222222222222222222222222222222222222222222222222222222</td></tr>
<tr><td>ChatGLM-CMExam</td><td>类风湿关节炎最常见的临床表现为关节肿痛,但除有关节肿痛外,对类风湿关节炎诊断最有意义的表现是足关节隆起部与受压部位皮下出现无痛性结节(B对)。足跟足掌部位痛(A错)可见于类风湿关节炎,但不是诊断最有意义的表现。弥漫性肺间质病变(C错)可见于肺结核,不属于对类风湿关节炎诊断最有意义的表现。胸腔积液(糖含量正常)(D错)可见于结核分枝杆菌感染,不属于对类风湿关节炎诊断最有意义的表现。小腿痛性皮下结节(E错)可见于风湿结节。</td><td>Rheumatoid arthritis most commonly presents with joint swelling and pain. However, in addition to joint swelling and pain, the most significant manifestation for diagnosing rheumatoid arthritis is the presence of painless nodules under the skin at the sites of joint prominence and pressure (B true). Pain in the heel and sole of the foot (A false) can be seen in rheumatoid arthritis, but it is not the most significant manifestation for diagnosis. Diffuse pulmonary interstitial disease (C false) can be seen in conditions such as pulmonary tuberculosis but is not the most significant manifestation for diagnosing rheumatoid arthritis. Pleural effusion (with normal glucose content) (D false) can be seen in infections caused by Mycobacterium tuberculosis but is not the most significant manifestation for diagnosing rheumatoid arthritis. Painful subcutaneous nodules in the lower leg (E false) can be seen in rheumatic nodules, but they are not the most significant manifestation for diagnosing rheumatoid arthritis.</td></tr>
<tr><td>LLaMA-CMExam</td><td>类风湿关节炎的主要病理表现为类风湿关节炎的关节隆起部与受压部位皮下出现无痛性结节（B对），关节隆起部皮下腔中有肉芽腺细胞增生，肉芽腺细胞增生后，肉芽腺细胞向肌内向外增大，形成肌内腔内的肉芽腺细胞细胞团，形成结节，并依赖结节细胞的肌肉细胞分泌</td><td>The main pathological manifestation of rheumatoid arthritis is the presence of painless nodules under the skin at the sites of joint prominence and pressure (B true). In the subcutaneous cavity of the joint prominence, there is proliferation of granulation tissue cells. After proliferation, these cells enlarge from the inside of the muscle outward, forming clusters of granulation tissue cells within the muscle cavity, which eventually develop into nodules. The formation of these nodules is dependent on the secretion of muscle cells by the nodular cells</td></tr>
<tr><td>Alpaca-CMExam</td><td>类风湿关节炎的临床表现：全身症状及体征；类风湿患者常有全身症状，如热病、痰热、腹痛、胸痛、肢体痛、足痛等，常有厌氧滋阴、腹泻、腹痛、肢体痛足痛等，病情急骤，严重者可出现腹痛、胸痛、足痛肢体痛、头痛、热病、痰热、腹痛、</td><td>Rheumatoid arthritis's clinical manifestations include systemic symptoms and signs. Rheumatoid patients often experience systemic symptoms such as fever, phlegm-heat, abdominal pain, chest pain, limb pain, and foot pain. They may also experience anorexia, diarrhea, abdominal pain, limb pain, and foot pain. The condition can worsen rapidly, and in severe cases, there might be abdominal pain, chest pain, foot pain, limb pain, headache, fever, and phlegm-heat,</td></tr>
<tr><td>Vicuna-CMExam</td><td>类风湿关节炎的主要病理变化为类风湿关节炎的类型及其发病程度。类风湿关节炎的病理变化主要包括：类风湿关节炎的类型及其发病程度。类风湿关节炎的病理变化主要包括：1.类风湿关节炎的类型及其发病程度；①类风湿关节炎的急性型；②类风湿关节炎的晚性型；③类风湿关节炎的慢性</td><td>The main pathological changes in rheumatoid arthritis are the types and severity of rheumatoid arthritis. The pathological changes in rheumatoid arthritis mainly include: 1. Types and severity of rheumatoid arthritis: ① Acute type of rheumatoid arthritis; ② Late type of rheumatoid arthritis; ③ Chronic type of rheumatoid arthritis</td></tr>
</table>

Figure 8: A case study of LLMs' generated explanations.

## A.4 Analysis of Model Generation Correctness

To assess the accuracy of model-generated explanations, we conducted a study using a randomly selected sample of 50 cases in which the Language Models (LLMs) correctly predicted the answers. Medical experts were then invited to manually verify the correctness of the explanations, focusing not only on the accuracy of the answer predictions but also on the quality of the accompanying explanations.

Our investigation revealed that despite the correct answer predictions by the models, certain samples exhibited errors in their corresponding explanations. These errors were categorized by the experts into three groups: explanations that were irrelevant, repeated, or inaccurate. The statistics presented in Figure 9 demonstrate that the number of samples with accurate explanations generated by the GPT models exceeded 45, accounting for over 90% of the total. However, it is important to note that both the ChatGLM and ChatGLM-CMExam models may produce some erroneous explanations, primarily consisting of inaccuracies and irrelevance. We have included examples of these incorrect explanations in Figure 10.

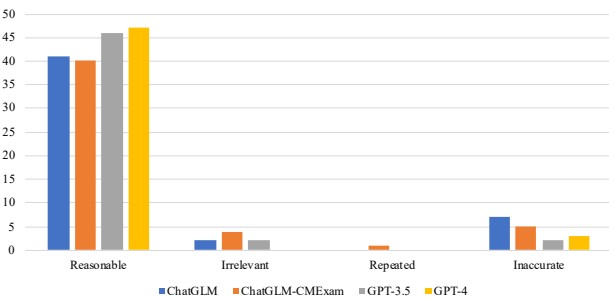

Figure 9: Correctness analysis.

## A.5 Analysis of Few-Shot and Chain-of-Thought Prompts

In our research, we designed few-shot and chain-of-thought prompts for the answer prediction and reasoning tasks and conducted experiments on the GPT models. As shown in Table 13, our results demonstrate that while the use of few-shot or chain-of-thought prompts did not yield significant improvements in the prediction task, but there was a notable enhancement in the reasoning task.

Specifically, for the GPT-4 model, the utilization of few-shot prompts increased the BLUE-1 from 0.17 to 5.95, and the BLUE-4 from 0.06 to 2.25. Furthermore, incorporating chain-of-thought prompts further increased the BLUE-1 to 7.29. Similarly, positive effects were observed on the GPT-3.5 model, where few-shot prompts improved the BLUE-1 and BLUE-4 to 14.62 and 4.80, respectively. Additionally, the ROUGE-1, ROUGE-2, and ROUGE-L increased to 38.08, 18.35, and 18.37.

These improvements can be attributed to the fact that few-shot prompts provide examples that GPT models can reference when generating detailed explanations for each option during the reasoning process. Similarly, chain-of-thought prompts can achieve similar effects, aiding in the enhancement of model performance.

Table 13: Few-shot and chain-of-thought prompting experiment results of GPT models

| Models | Prediction | | Reasoning | | | | |
|---|---|---|---|---|---|---|---|
| | ACC | f1 | BLUE-1 | BLUE-4 | ROUGE-1 | ROUGE-2 | ROUGE-L |
| GPT-4 | 61.6%±0.1 | 61.7%±0.1 | 0.17±0.00 | 0.06±0.00 | 29.74±0.09 | 14.84±0.04 | 11.51±0.03 |
| GPT-4_few-shot | 62.0%±0.4 | 61.4%±0.5 | 5.95±0.12 | 2.25±0.07 | 37.24±0.35 | 19.23±0.26 | 17.24±0.07 |
| GPT-4_cot | 61.6%±0.9 | 61.4%±0.9 | 7.29±0.71 | 2.20±0.25 | 35.85±0.78 | 16.79±0.83 | 17.18±0.30 |
| GPT-3.5 | 46.4%±0.0 | 46.2%±0.1 | 3.56±0.08 | 1.49±0.06 | 33.80±0.11 | 16.39±0.05 | 14.83±0.13 |
| GPT-3.5_few-shot | 45.3%±0.6 | 44.9%±0.6 | 14.62±0.16 | 4.80±0.06 | 38.08±0.44 | 18.35±0.16 | 18.37±0.29 |
| GPT-3.5_cot | 47.9%±0.7 | 47.7%±0.7 | 13.47±0.52 | 3.69±0.18 | 36.47±0.42 | 16.41±0.24 | 17.82±0.31 |

| Question | Answer | Explanation | Model-generated explanation | Error classification | Expert analysis |
|---|---|---|---|---|---|
| 女，40岁，半月前头痛、恶心，呕吐2次，伴低热，向话不答，独自发呆，眼肌床不起，抽搐2次，血压140/90mmHg，下肢肌床力增粗，右侧巴氏征（+）。脑电图：弥漫性波以额颞为主；有待发生尖波发放。首选的检查方法是：
A. 脑MRI
B. 脑CT
C. 心电图
D. 脑脊液
E. 智能测试

Female, 40 years old. Half a month ago, she experienced headaches, nausea, and vomiting twice, accompanied by low-grade fever, unresponsiveness to questioning. She stared blankly, stayed in bed without getting up, and had two episodes of convulsions. Blood pressure 140/90 mmHg, increased muscle tone in the lower limbs, positive Babinski sign on the right side. Electroencephalogram (EEG): Diffuse delta waves predominant in the frontal and cervical regions, with occasional spike wave discharges. The preferred diagnostic method is:
A. Brain MRI
B. Brain CT
C. Electrocardiogram (ECG)
D. Cerebrospinal fluid analysis
E. Cognitive testing | A | **CHATGLM:** 根据病人的症状和身体检查结果，首选检查方法是脑MRI，脑MRI可以显示脑组织的形态和功能，特别是在处理脑电图结果时非常有用。此外，脑MRI还可以帮助我们确定与头痛和恶心的原因。这对于治疗和康复是非常重要。

**CHATGLM:** Based on the patient's symptoms and examination results, the preferred diagnostic method is brain MRI. Brain MRI can provide visualization of brain morphology and function, making it particularly useful when interpreting EEG results. Additionally, brain MRI can help determine the underlying causes of headaches and nausea, which is crucial for treatment and | Irrelevant

Irrelevant | 对MRI作用的解释无关

The explanation of the effect of MRI is irrelevant. |
| | A | 该患者半月前头痛、恶心，呕吐2次，伴体，下肢肌床力增粗性，右侧发性尖波发放，应首选脑MRI（A对），MRI可发现T1低信号、T2高信号病灶，从而对低下弥漫和恶心的原因。与CT比较，更能准确地找出发病初期的变化。

本题考查不能纳入基本医疗保险用药范围的。C栏规则正确，以下药品不能纳入基本医疗保险用药范围：（一）（各类营养滋补药品）；（二）部分可以入药的动物及动物脏器，干（水）果类；（三）中药材和中药饮片中的各种酒剂；（四）各类药品中的果味制剂、泡腾剂；（五）血液制品、蛋白类制品（特殊适应症与急救、抢救除外）；（六）劳动保障规定基本医疗保险基金不予支付的其他药品。 | **CHATGLM-M:** 本题考查的是能纳入基本医疗保险用药范围。不能纳入基本医疗保险用药范围的是果味制剂（E对）。

The patient had a history of headaches, nausea, and unresponsiveness (manifested as reduced speech and psychomotor inhibition). Examination revealed increased muscle tone in the lower limbs and positive Babinski sign on the right side. The EEG showed diffuse delta waves predominant in the frontal and cervical regions, with occasional spike wave discharges. Therefore, the diagnosis is psychiatric disorder due to viral encephalitis. The preferred diagnostic method is brain MRI (A true), as MRI can detect T1 hypointense and T2 hyperintense brain parenchymal lesions. Compared to CT, MRI is more accurate in identifying early changes (B false) and can guide appropriate treatment. | | |
| 不能纳入基本医疗保险用药范围的是：
A. 中药材
B. 中药饮片
C. 中成药 （特殊适应症）
D. 血液制品
E. 果味制剂

The options that are not covered under basic medical insurance for medication are:
A. Chinese herbal medicine
B. Chinese herbal tea pills
C. Traditional Chinese patent medicine
D. Blood products (special indications)
E. Fruit-flavored preparations | E | This question examines the medications that are not covered under basic medical insurance. The option that is not covered under basic medical insurance for medication is fruit-flavored preparations (Option E). According to the "Interim Measures for the Administration of the Scope of Basic Medical Insurance for Urban Employees," Article 4 states that the following drugs are not covered under basic medical insurance: (1) Drugs primarily used for nutritional supplementation; (2) Some animal and animal organ products that can be used as medicine, dried (water) fruits; (3) Various wine preparations made from Chinese herbal materials and Chinese herbal tea pills; (4) Fruit-flavored preparations and effervescent tablets in various medications; (5) Blood products, protein products (except for special indications); (6) Other drugs not covered by the basic medical insurance fund as stipulated by the Ministry of Labor Security. | **ChatGLM-CMExam:** 本题考查的是医保药品目录制定。不能纳入基本医疗保险用药范围的是果味制剂（E对）。国家医疗保障局负责制定国家基本医疗保险药品目录和临床必需药品两大类。临床必需药品是指符合下列条件之一的药品：治疗：◇:◇:◇:◇:◇:◇:◇:◇:◇:◇:◇:◇:◇:◇:◇:◇:◇:◇:◇:◇:◇:◇:◇:◇:◇:◇:◇:◇:◇:◇:◇:◇:◇:◇:◇:◇:◇:◇:◇:◇:◇:◇:◇:◇:◇:◇:◇:◇:◇:◇:◇:◇:◇:

**ChatGLM-CMExam:** This question examines the establishment of the medical insurance drug list. The option that is not covered under basic medical insurance is fruit-flavored preparations (Option E). The formulation of the medical insurance drug list is the responsibility of the National Healthcare Security Administration, and it is divided into two categories: clinically necessary drugs and non-clinically necessary drugs. Clinically necessary drugs refer to drugs that meet one of the following conditions: treatment: ◇:◇:◇:◇:◇:◇:◇:◇:◇:◇:◇:◇:◇:◇:◇:◇:◇:◇:◇:◇:◇:◇:◇:◇:◇:◇:◇:◇:◇:◇:◇:◇:◇:◇:◇:◇:◇:◇: | Repeated

Repeated | 轻易可见重复现象

Repeated phenomena can be easily observed. |
| 导致代谢性碱中毒的原因是：
A. 颅脑损伤
B. 大量胸腔积液
C. 喉头水肿
D. 低钾血症
E. 异物阻塞气管

The causes of metabolic alkalosis are:
A. Cranial brain injury
B. Large amount of pleural effusion
C. Laryngeal edema
D. Hypokalemia
E. Tracheal obstruction by a foreign body | D | 低钾血症（D对）时引起细胞外液K+浓度降低，引起细胞内K向细胞外移动，同时细胞外H+移入细胞，可发生代谢性碱中毒。颅脑损伤（A错）、喉头水肿（C错）、异物阻塞气管（E错）都可以使通气障碍，引起呼吸性酸中毒。大量胸腔积液（B错）大量胸腔积液引起通气障碍而受阻而引起呼吸性酸中毒。

Hypokalemia (D true) leads to a decrease in extracellular potassium concentration, causing intracellular potassium to move out of the cells. At the same time, extracellular hydrogen ions (H+) move into the cells, leading to metabolic alkalosis. Cranial brain injury (A false), large amount of pleural effusion (B false), laryngeal edema (C false), and tracheal obstruction by a foreign body (E false) can all cause respiratory acidosis due to impaired ventilation and the consequent obstruction of a foreign body (E false) can all potentially lead to respiratory acidosis. | **ChatGLM-CMExam:** 低钾血症时细胞外液pH降低，细胞外液pH降低，导致代谢性碱中毒（D对）、颅脑损伤（B错）、颅脑损伤致CO2排出过多而引起呼吸性碱中毒。

**ChatGLM-CMExam:** In cases of hypokalemia, the extracellular potassium concentration decreases, resulting in a decrease in extracellular pH, leading to metabolic alkalosis (D true), Cranial brain injury (A false), large amount of pleural effusion (B false), laryngeal edema (C false), and tracheal obstruction by a foreign body (E false) can all potentially lead to respiratory acidosis due to impaired ventilation and the consequent obstruction of a foreign body (E false) can all potentially lead to respiratory acidosis due to impaired CO2 elimination. | Inaccurate

Inaccurate | 细胞外液pH升高

The extracellular pH increases |

Figure 10: Examples of incorrect model-generated explanation.

## A.6 Data statistics

Questions in CMExam have a median length of 17 (Q1: 12, Q3: 32). Regarding solution explanations, the median length is 146 tokens (Q1: 69, Q3: 247). Table 14 shows more basic statistics of CMExam,

Table 14: Basic statistics of CMExam. Q: questions; E: explanations; Q1/3: the first/ third quantile.

|  | Train | Dev | Test | Total |
|---|---|---|---|---|
| Question # | 54,497 | 6,811 | 6,811 | 68,119 |
| Vocab | 4,545 | 3,620 | 3,599 | 4,629 |
| Max Q tokens | 676 | 500 | 585 | 676 |
| Max E tokens | 2,999 | 2,678 | 2,680 | 2,999 |
| Avg Q tokens | 29.78 | 30.07 | 32.63 | 30.83 |
| Avg E tokens | 186.24 | 188.95 | 201.44 | 192.21 |
| Median (Q1, Q3) Q tokens | 17 (12, 32) | 18 (12, 32) | 18 (12, 37) | 18 (12, 32) |
| Median (Q1, Q3) E tokens | 146 (69, 246) | 143 (65, 247) | 158 (80, 263) | 146 (69, 247) |

## A.7 Guidelines for Expert-Annotation

During the annotation phase, we invited one expert physician from the Second Affiliated Hospital of Zhejiang University and one senior doctoral student from Zhejiang University School of Medicine to carry out the annotations. The expert physician has over two years of clinical experience. The annotation guidelines have the following sections:

1. Comprehensive Question Understanding: Prior to initiating the annotation process, meticulously comprehend the medical question, ensuring a holistic grasp of its context and significance.

2. Subject Categorization: Identify the precise subject or medical field that the question pertains to, such as cardiology, pediatrics, or pathology.

3. Principal Symptoms or Medical Conditions: Ascertain and pinpoint the primary symptoms or medical conditions expounded in the question.

4. Examination of Pertinent Factors: Scrutinize the question for any associated factors that might be present, including the severity of the ailment, its etiology, and patient history given in the question.

5. Examination of Pertinent Factors: Scrutinize the question for any associated factors that might be present, including the severity of the ailment, its etiology, and patient history given in the question.

6. Appropriate Classification System Usage: Use the accurate classification system for annotation in alignment with the determined subject and symptoms. Suitable systems could encompass the 11th revision of the International Classification of Diseases (ICD-11), the Directory of Medical Institution Diagnostic and Therapeutic Categories (DMIDTC), and others.

7. Addressing Multiple Annotations: In scenarios where the question encompasses multiple symptoms or medical conditions, opt for the most related classification for annotation.

8. Ensuring High-Quality Annotations: Adhere to the guidelines and definitions within the chosen classification system. This diligence helps avert subjectivity and ambiguity, fostering precision in the annotations.

9. Navigating Queries and Uncertainties: Should any doubts or uncertainties emerge during the annotation process, consult the official documents and glossaries of the chosen classification system. Engaging in discussions with professionals is also advised to achieve clarity.

10. Resolving Discrepancies: When disagreements emerge between annotators, a collaborative discussion shall be initiated. The objective is to reach a consensus and unify the annotation decision.

## A.8 Prompt strategies for Pre-Annotation

During the experimental process, we indeed tried different prompts to enable GPT to better understand and complete the annotation task. The specific strategies were as follows:

1. Without ICD-11 Category Instructions: We did not provide detailed ICD-11 category information as instruction but directly supplied the question information and asked GPT to respond. Under this setup, a significant portion of the categories returned by GPT did not strictly belong to ICD-11 classifications, yielding unsatisfactory results.

2. Batch Processing for Cost Efficiency: Initially, we concatenated multiple questions and, through a single dialogue, had GPT return annotations for multiple questions. Under this setup, expert validation showed that the accuracy of GPT's annotations was relatively low.

3. Consistency in Formatting: When no format guidance was given, GPT's return format was inconsistent, resulting in a higher parsing cost. Hence, after multiple trials, we eventually opted for more rigorous format guidance.

   Our annotation process was carried out in two stages: First, GPT conducted an initial round of pre-annotation. Subsequently, we invited an expert physician from the Second Affiliated Hospital of Zhejiang University and a doctoral student from Zhejiang University School of Medicine to annotate. The expert physician had over two years of clinical experience. In instances where there were disagreements in annotations, both annotators would discuss and eventually arrive at a consensus.

