# OpenReview forum: "Benchmarking Large Language Models on CMExam - A comprehensive Chinese Medical Exam Dataset"
_NeurIPS.cc/2023/Track/Datasets_and_Benchmarks — NeurIPS 2023 Datasets and Benchmarks Poster_

### Official Review · Reviewer_Vnjt · 2023-07-20
**QA Benchmark based sourced from Chinese National Medical Licensing Examination**

**Rating:** 7
**Confidence:** 3
**Clarity:** The paper is well written

**Strengths:**

An important contribution of the paper is the introduction of furhter classification for each question in the dataset.
This allows to stratify results along multiple axis, e.g. according to medical competency.
Second major contribution is the extensive benchmarking performed. The authors assessed the performance of 11 LLMs,
fine-tuning 5 on their dataset.

**Additional Feedback:**

You did a terrific job with the benchmarking.
I understand that the benchmark is tailored towards assessing the "intrinsic" capabilities of LLMs.
However one class of model not considered was a "retriever + reader architecture".
That is, a system which uses e.g. BM25 to retrieve relevant document(s) for a given query from e.g. the Chinese Wikipedia Dump.
The query and the document(s) are then passed to e.g. a BERT model to select the answer.

If possible, authors should release explanations generated from different models. It would be an interesting read.

**Correctness:**

The dataset is constructed appropriately and the benchmarking follows standard evaluation protocol for the task.

**Documentation:**

The dataset is released as a github repository.
Ethical and responsible use and intended uses are reported in the paper but not reported in the dataset webpage.
No maintainance plan is reported.
Code for reproducing the benchmark is missing but listed as a TODO (planned?).

**Ethics:**

I do not see any ethical concerns with the submission..

**Limitations:**

The authors appropriately addressed limitations and potential negative societal impact of their work.

**Opportunities For Improvement:**

As mentioned in the related work, there exist already a QA dataset derived from Chinese National Medical Licensing Examination:
MLEC-QA. The authors already specify that their dataset includes further anntoations for each question.
I find however that this may not be enough as novel aspect.
I suggest to include a table stressing every aspect in which CMExam differs from MLEC-QA to shine a light
on the contributions speficic to the dataset.

The second remark is about the  "innovative strategy called GPT-Assisted Annotation".
This is mentioned in the introduction and not discussed only briefly in the appendix.
As this was used to help human annotators in the classifications of question,
the major novelty of the dataset, a more elaborate description is necessary.
LLMs are highly sensitive to difference prompts (https://aclanthology.org/2021.acl-long.295/). How do you deal with this?
How many rounds of annotations? What was the inter-annotator agreement?

Finally, the authors mention that "85.24% of questions have brief solution" explanations.
How were this explanations generated? Are they already provided in the Chinese National Medical Licensing Examination.

**Relation To Prior Work:**

Another dataset worth to be mentioned:
- Meddialog: https://arxiv.org/abs/2004.03329


Recent work (published relatively shortly before the submission deadline: https://arxiv.org/abs/2303.13375v1, See Section 3.1.1)
specifically probes GPT-4 on question requiring images without providing them as input.
This finding should be cited.

**Summary And Contributions:**

The authors introduce a multiple-choice QA benchmark sourced from the  Chinese National Medical Licensing Examination.
Each question is further annotated by the authors along 5 axis: ICD coding, clinical department, medical
disciplines, primary medical competency and difficulty.
The annotation is carried by humans, initial pre-annotation was done by GPT-4.
The authors proceed to perform an extensive benchmarking with multiple general domain LLMs (e.g. GPT-4) and
domain specific ones, e.g. DoctorGLM and a fine-tuned LLaMA.
They find that  GPT-4 shows strong zero-shot performance and further stratify results analysis along the 5 classifications introduced for each question.

---

> ### Author Response · Authors · 2023-08-19
> **Comments for Q1, Q2 and Q3**
>
> Dear Reviewer Vnjt,
>
> We appreciate your thorough examination of our manuscript, highlighting areas of concern, and providing valuable insights to enhance our study. We have meticulously reviewed all comments and addressed each question and suggestion. We are confident that these changes have strengthened our paper. If there are any additional questions, please do not hesitate to let us know.
>
> > **Q1**: As mentioned in the related work, there exist already a QA dataset derived from Chinese National Medical Licensing Examination: MLEC-QA. The authors already specify that their dataset includes further anntoations for each question. I find however that this may not be enough as novel aspect. I suggest to include a table stressing every aspect in which CMExam differs from MLEC-QA to shine a light on the contributions speficic to the dataset.
>
> **A1**: We acknowledge the importance of highlighting the novel aspects of CMExam in comparison to MLEC-QA. To this end, we have created a table to demonstrate the key differences between MLEC-QA and our dataset. CMExam was designed to serve as a benchmark for LLM performance evaluation in the medical field. It distinguishes itself from MLEC-QA through annotations with authoritative references, difficulty ratings, and question explanations. Our contribution also extends beyond the dataset itself. In addition to providing CMExam, we emphasize a comprehensive benchmark of LLM performance on CMExam, a facet absent from MLEC-QA, which only compared 3 non-machine learning algorithms, 3 BERT variants, and 2 RoBERTa variants. Detailed comparison results have been updated to the following GitHub link: https://github.com/williamliujl/CMExam/blob/main/docs/Comparison_with_MLEC-QA.jpg
>
> While we don't intend to include this table directly in our manuscript at this point due to its potential narrow focus on a specific paper within our study, we have revised our related work description to encompass a more comprehensive exploration of the dissimilarities between the two datasets based on the table to underscore our distinctive contribution
>
> > **Q2**: The second remark is about the "innovative strategy called GPT-Assisted Annotation". This is mentioned in the introduction and not discussed only briefly in the appendix. As this was used to help human annotators in the classifications of question, the major novelty of the dataset, a more elaborate description is necessary. LLMs are highly sensitive to difference prompts (https://aclanthology.org/2021.acl-long.295/). How do you deal with this? How many rounds of annotations? What was the inter-annotator agreement?
>
> **A2**: Thank you for your suggestion. We agree that this is an innovative aspect of our research and we have included related information in the updated version. Below are answers to your questions:
>
> During the experimental process, we indeed tried different prompts to enable GPT to better understand and complete the annotation task. The specific strategies were as follows:
> * Without ICD-11 Category Instructions: We did not provide detailed ICD-11 category information as instruction but directly supplied the question information and asked GPT to respond. Under this setup, a significant portion of the categories returned by GPT did not strictly belong to ICD-11 classifications, yielding unsatisfactory results.
> * Batch Processing for Cost Efficiency: Initially, we concatenated multiple questions and, through a single dialogue, had GPT return annotations for multiple questions. Under this setup, expert validation showed that the accuracy of GPT's annotations was relatively low.
> * Consistency in Formatting: When no format guidance was given, GPT's return format was inconsistent, resulting in a higher parsing cost. Hence, after multiple trials, we eventually opted for more rigorous format guidance.
>
> Our annotation process was carried out in two stages: First, GPT conducted an initial round of pre-annotation. Subsequently, we invited an expert physician from the Second Affiliated Hospital of Zhejiang University and a doctoral student from Zhejiang University School of Medicine to annotate. The expert physician had over two years of clinical experience. In instances where there were disagreements in annotations, both annotators would discuss and eventually arrive at a consensus.
>
> > **Q3**: Finally, the authors mention that "85.24% of questions have brief solution" explanations. How were this explanations generated? Are they already provided in the Chinese National Medical Licensing Examination.
>
> **A3**: The Chinese National Medical Licensing Examination does not provide explanations. After collecting the question data from the internet, we manually matched these questions with those found in multiple online medical question-answering platforms to obtain the corresponding explanations for the matched questions.

---

> > ### Comment · Reviewer_Vnjt · 2023-08-25
> >
> > Thank you for addressing my comments.
> >
> > Altough, as I alreayd said, the authors did a terrific work by benchmarking all those models the major contribution still is the dataset (we cannot "blame" previous work for not trying out approaches that were not available at the time).
> >
> > CMExam introduces **signicant** novel annotation layers if compared with previous work. It is just not easy to see it from the paper.
> > For instance:
> >
> > > A3: The Chinese National Medical Licensing Examination does not provide explanations. After collecting the question data from the internet, we manually matched these questions with those found in multiple online medical question-answering platforms to obtain the corresponding explanations for the matched questions.
> >
> > This is a novel contribution involving a laborious effort by the annotators. Yet nowhere in the paper it is mentioned that this is something the authors introduced de novo.
> >
> > The way I see it Table 2 can easily go in the appendix. It should be replaced with a more informative table describing the novel annotation layers. Something similar to this one (without the comparison with MLEC-QA)
> >
> > > GitHub link: https://github.com/williamliujl/CMExam/blob/main/docs/Comparison_with_MLEC-QA.jpg
> >
> > The table should also include a column specifying how the "pre-annotations" were generated with accompanying text in the Appendix
> > with a detailed description.
> > E.g. Did the authors use GPT-4 also to pre-annotate the question with  DMIDTC? And the " List of Graduate Education Disciplinary Majors" too? Was the  prompt always the same?
> > In the case of explanations, which "online medical QA platforms"?
> >
> > And maybe, for clarity, a column with an example annotation, e.g.
> >
> > > The fourth dimension  was created by two medical professionals within the team to assess the primary medical competency140
> > tested by each associated question. It consists of four categories.
> >
> > What are these four categories?
> >
> > Finally, the authors should provide an easier access to the annotations in their released dataset.
> > I do not know whether this is becasue I cannot read Chinese but in the sharded CSV files I cannot see
> > for instance the associated ICD-11 code of each question.
> > I tried running the notebook for the dataset statistics:
> >
> > https://github.com/williamliujl/CMExam/blob/main/src/preprocess/data%20stats.ipynb
> >
> > but with no success. The following code throws an error complaining that the 'ICD-11' column does not exists:
> > ```python
> > df = pd.concat([pd.read_csv("data/train.csv"),  pd.read_csv("data/val.csv")])
> > df["ICD-11"].value_counts()["不符合"]
> > ```
> > If these concerns are addressed I am more than willing to raise my score.

---

> > > ### Author Response · Authors · 2023-08-28
> > > **Comments for Q1, Q2, Q3, Q4 and Q5**
> > >
> > > Thank you for the detailed comment. We've categorized your concerns and made the appropriate revisions, as summarized below:
> > >
> > > > **Q1**: Major Contribution and Novel Annotation Layers
> > >
> > > **A1**: Thank you for explicitly pointing out our omissions. We did not realize how important this step was, and we have added descriptions of the solutions to the data characteristics section.
> > >
> > > `Despite the inherent absence of explanations within the CNMLE, we cross-referenced exam questions with solutions offered by diverse online medical examination preparation platforms, effectively enhancing the dataset's informational depth. `
> > >
> > > >**Q2**: Table 2 and Suggested Replacements
> > >
> > > **A2**: Thank you for your further instructions. We agree that Table 2 can go to the appendix. We have made a new Table 2 which reflects the innovations of our dataset. Below shows the new table:
> > >
> > > |     Annotation Content     |          References         | Unique values |
> > > |----------------------------|-----------------------------|---------------|
> > > |       Disease Groups       | The 11th revision of ICD-11 |      27       |
> > > |   Clinical Departments     | The Directory of Medical Institution Diagnostic and Therapeutic Categories (DMIDTC) | 36 |
> > > |    Medical Disciplines     | List of Graduate Education Disciplinary Majors (2022) | 7 |
> > > |   Medical Competencies     |    Medical Professionals    |       4       |
> > > |      Difficulty Level      |       Human Performance     |       5       |
> > >
> > > This new table 2 is now introduced right after mentioning CMExam in related work, specifically:
> > >
> > > `[...] These features make CMExam an indispensable resource [...]. Table 2 presents a list of innovations and characteristics of CMExam, which are discussed in detail in the following sections.`
> > >
> > > >**Q3**: More details of the pre-annotation part
> > >
> > > **A3**:
> > >
> > > For your questions:
> > >
> > > 1. Did the authors use GPT-4 also to pre-annotate the question with DMIDTC?
> > >
> > >     Yes.
> > >
> > > 2. The "List of Graduate Education Disciplinary Majors" too?
> > >
> > >     Yes.
> > >
> > > 3. Was the prompt always the same?
> > >
> > >     Yes.
> > >
> > > 4. In the case of explanations, which "online medical QA platforms"?
> > >
> > >     医考帮 (Yikaobang), 金英杰医学 (Jinyingjie), 蓝基因 (Lanjiyin).
> > >
> > > >**Q4**: Example annotations.
> > >
> > > **A4**: Since pre-annotation examples are too lengthy to fit in table cells, we made them figures in the appendix (supplementary figures 1-4).
> > >
> > > >**Q5**: What the four categories are for medical competency
> > >
> > > **A5**: The categories can be found in supplementary Table 4 (shown below), as we mentioned shortly after the highlighted sentence: "For detailed information on these additional annotations, please refer to supplementary Tables 1 to 4." In case the reviewer and readers were distracted by the listed values of the difficulty dimension, we removed the unique values listed and added more detailed instructions in the paper:
> > >
> > > `The fifth dimension represents five potential difficulty levels of each question, determined by analyzing the correctness rate observed in human performance data collected alongside the questions. For detailed information on these additional annotations including their potential values, please refer to supplementary Tables 1 to 4.`
> > >
> > > In case the reviewer is simply asking what the four categories in medical competency are:
> > >
> > > | Abbreviation | Full English Name                            | Chinese Name |
> > > |--------------|----------------------------------------------|--------------|
> > > | Diag         | Disease Diagnosis and Differential Diagnosis | 疾病诊断和鉴别诊断    |
> > > | MedFund      | Medical Fundamentals                         | 医学基础知识       |
> > > | N/A          | Not Applicable                               | 不符合          |
> > > | PHL          | Public Health Law and Ethics                 | 公共卫生法律伦理     |
> > > | Treat        | Disease Treatment                            | 疾病治疗         |

---

> > > ### Author Response · Authors · 2023-08-28
> > > **Comments for Q6 and Q7**
> > >
> > > > **Q6**: Access to Annotations in the Released Dataset
> > >
> > > **A6**: We mentioned that annotations are available only in the test set. The reference in our manuscript shows as: "[...] questions in the test dataset contain additional annotations". If you open the file "test.csv," (now renamed to test_with_annotations.csv for clarity) you will see the header:
> > > "Question, Options, Answer, Explanation, Disease Group, Area of Competency, Clinical Department, Medical Discipline, Difficulty level."
> > >
> > > If the readers are interested in screening the data but do not read Chinese, they can refer to 1. the data stats ipynb, which contains dictionaries of translations, and 2. the supplementary tables 1-4 for term-wise translations.
> > >
> > > In addition, the reviewer mentioned they did not see ICD-11 codes. We did not use ICD-11 codes but the labels for clarity. They can be easily converted using the two measures we mentioned above, or, simply via the official references: https://icd.who.int/browse11 and https://icd.who.int/browse11/l-m/zh. One example would be 02 (code) - Neo (abbreviation) - Neoplasms (label in English) - 肿瘤 (label in Chinese, found in test_with_annotations.csv). In case some readers are used to working with ICD codes, we updated Supplementary table 1 with ICD-11 codes.
> > >
> > > >**Q7**: Problems in the current data statistics .ipynb.
> > >
> > > **A7**: The code threw an error because you didn't include the test.csv file. For clarity, we have renamed the test.csv file to test_with_annotations.csv. We acknowledge that the original script contains code we used to tidy up the dataset (e.g., "ICD-11" was renamed to "Disease Group"). We have also restructured it and removed code no longer needed, so the readers should be able to just run it through now. Thank you for spotting the issue.

---

> > > > ### Comment · Reviewer_Vnjt · 2023-08-29
> > > >
> > > > Thank you for addressing all my comments.
> > > >
> > > > >4.  In the case of explanations, which "online medical QA platforms"?
> > > >
> > > > > 医考帮 (Yikaobang), 金英杰医学 (Jinyingjie), 蓝基因 (Lanjiyin).
> > > >
> > > > Please add these with links to the main text (footnote).
> > > > This is a dataset paper and as such it must provide as much details as possible on how it was contsructed.
> > > >
> > > > > The code threw an error because you didn't include the test.csv file
> > > >
> > > > Does this mean that the annotations (ICD-11, DMIDTC, ...) are **only** in the test set?
> > > > If so please make it clear in the paper.

---

> > > > > ### Author Response · Authors · 2023-08-29
> > > > > **Comments for Q1 and Q2**
> > > > >
> > > > > Thank you for the detailed comment. We've categorized your concerns and made the appropriate revisions, as summarized below:
> > > > >
> > > > > >**Q1**: Please add these with links to the main text (footnote). This is a dataset paper and as such it must provide as much details as possible on how it was contsructed.
> > > > >
> > > > > **A1**: Thank you for your further instructions. We have added the information to the main text.
> > > > >
> > > > > >**Q2**: Does this mean that the annotations (ICD-11, DMIDTC, ...) are only in the test set? If so please make it clear in the paper.
> > > > >
> > > > > **A2**: Yes, you are correct. Due to the significant costs associated with annotations, we have currently only annotated the test set. We have mentioned this in the main text of the paper, please refer to the subsection "Data Collection and Pre-processing".
> > > > >
> > > > > ` Each question in the dataset is associated with an ID, five candidate answers, and a correct answer. 85.24% of questions have brief solution explanations and questions in the test set contain additional annotations.`
> > > > >
> > > > > In future work, we plan to extend annotations to both the training and validation sets.

---

> > > > > ### Author Response · Authors · 2023-08-30
> > > > > **Look forward to further feedback**
> > > > >
> > > > > Dear Reviewer Vnjt:
> > > > >
> > > > > We have taken into account all of your concerns and sincerely hope to earn your endorsement. We greatly appreciate your opinions and feedback, which is why we value your reconsideration of our paper.
> > > > >
> > > > > We would like to reiterate the importance of our endeavor in introducing CMExam, which consists of 60K+ multiple-choice questions and five additional question-wise annotations, and conducting thorough experiments with representative LLMs and QA algorithms on CMExam. Our deliverables hold the potential to provide the medical community with valuable insights.
> > > > >
> > > > > As the discussion period draws to a close, we kindly and respectfully ask you to consider **raising the rating of our work**. Thank you for your time and consideration.
> > > > >
> > > > > Best,
> > > > >
> > > > > Authors of Paper425

---

> > > > > > ### Comment · Reviewer_Vnjt · 2023-08-30
> > > > > >
> > > > > > > A2: Yes, you are correct. Due to the significant costs associated with annotations, we have currently only annotated the test set. We have mentioned this in the main text of the paper, please refer to the subsection "Data Collection and Pre-processing".
> > > > > >
> > > > > > Sorry I missed that.
> > > > > >
> > > > > > > Thank you for your further instructions. We have added the information to the main text.
> > > > > >
> > > > > > Last minor remark. The footnote with the online sources for questions should go in "Dataset Characteristics" otherwise it's hard to understand what they are.
> > > > > >
> > > > > > I have raised my score.

---

> > > > > > > ### Author Response · Authors · 2023-08-30
> > > > > > > **Appreciate your feedback and endorsement of our work**
> > > > > > >
> > > > > > > Dear Reviewer Vnjt:
> > > > > > >
> > > > > > > Thank you for your recommendation regarding footnotes. We will take it into consideration and make the necessary updates in future versions of the paper.
> > > > > > >
> > > > > > > We greatly appreciate your feedback and endorsement of our work.
> > > > > > >
> > > > > > > Best,
> > > > > > >
> > > > > > > Authors of Paper425

---

> ### Author Response · Authors · 2023-08-19
> **Comments for Q4, Q5, Q6 and Q7**
>
> >**Q4**: Another dataset worth to be mentioned: Meddialog: https://arxiv.org/abs/2004.03329
>
> **A4**：Thank you for bringing up the dataset. The mentioned dataset and related discussion has been added to our updated manuscript.
>
> >**Q5**: Recent work (published relatively shortly before the submission deadline: https://arxiv.org/abs/2303.13375v1, See Section 3.1.1) specifically probes GPT-4 on question requiring images without providing them as input. This finding should be cited.
>
> **A5**: Thank you for bringing up the related work. We have included the mentioned citation and expanded the discussion in the updated version.
>
> >**Q6**: Ethical and responsible use and intended uses are reported in the paper but not reported in the dataset webpage. No maintainance plan is reported. Code for reproducing the benchmark is missing but listed as a TODO (planned?).
>
> **A6**: Thank you for bringing this to our attention. We have outlined the maintenance plan in Appendix A.7.7. All the required code and comprehensive guidelines have been uploaded to our GitHub repository: https://github.com/williamliujl/CMExam.
>
> **Who will  be supporting/hosting/maintaining the dataset?**
>
> Junling Liu, the corresponding author.
>
> **How can the owner/curator/manager of the dataset be contacted (e.g., email address)?**
>
> Via email: william.liuj@gmail.com
>
> **Is there an erratum? If so, please provide a link or other access point.**
>
>  No.
>
> **Will the dataset be updated (e.g., to correct labeling errors, add new instances, delete instances)? If so, please describe how often, by whom, and how updates will be communicated to dataset consumers (e.g., mailing list, GitHub)?**
>
> No, currently there are no scheduled updates for the dataset. If any updates are contemplated in the future, we will provide a clear explanation on the GitHub repository.
>
> **If the dataset relates to people, are there applicable limits on the retention of the data associated with the instances (e.g., were the individuals in question told that their data would be retained for a fixed period of time and then deleted)? If so, please describe these limits and explain how they will be enforced.**
>
> Not applicable.
>
> **Will older versions of the dataset continue to be supported/hosted/maintained? If so, please describe how. If not, please describe how its obsolescence will be communicated to dataset consumers.**
>
> The ongoing support and maintenance of older dataset versions will be contingent upon the identification of errors or inaccuracies. Given the nature of this historical medical exam dataset, the concept of obsolescence does not inherently apply. Therefore, consistent updates are not anticipated unless critical discrepancies are identified and addressed.
>
> **If others want to extend/augment/build on/contribute to the dataset, is there a mechanism for them to do so? If so, please provide a description. Will these contributions be validated/verified? If so, please describe how. If not, why not? Is there a process for communicating/distributing these contributions to dataset consumers? If so, please provide a description.**
>
> In the current version, we are unable to accommodate external merges or push requests for the dataset on GitHub due to the time-intensive nature of maintaining quality control over additional annotations. However, we do welcome submissions of new Chinese medical licensing exam questions without annotations. Such new questions can serve as unlabeled test sets. For those interested in expanding the dataset by incorporating new annotations, we have designed the dataset structure and annotation guidelines to facilitate this process. The guidelines can be found in the updated version of our paper. If substantial contributions result in an extended dataset, it could be released as "CMEXAM version 2" in subsequent research articles.
>
> >**Q7**: You did a terrific job with the benchmarking. I understand that the benchmark is tailored towards assessing the "intrinsic" capabilities of LLMs. However one class of model not considered was a "retriever + reader architecture". That is, a system which uses e.g. BM25 to retrieve relevant document(s) for a given query from e.g. the Chinese Wikipedia Dump. The query and the document(s) are then passed to e.g. a BERT model to select the answer.
>
> **A7**: Thank you very much for your appreciation. In this paper, our main focus lies on end-to-end models, where we benchmark several well-known existing models of the class. This comparison style is also observed in most pre-trained model benchmarking papers, especially for large lanague models beyond BERT.
> However, we are working on implementing the retriever + reader systems you mentioned and comparing them with end-to-end models in a seperate work. If you're interested, please keep an eye out for updates :)

---

> ### Author Response · Authors · 2023-08-19
> **Comments for Q8**
>
> > **Q8**: If possible, authors should release explanations generated from different models. It would be an interesting read.
>
> **A8**:  Thank you for your interest. We already reported examples of generated texts from all models we tested in Appendix - Figure.2. We also analyzed the results (analysis attached below). If you have any further questions or require additional information, please let us know.
>
> "In Figure 2, we present partial explanations generated by various models for a medical question from the CMExam dataset. Notably, GPT-4 and GPT-3.5 produce concise and sensible explanations, which may account for the lower BLEU scores. Conversely, models like Vicuna, LLaMA, and Huotuo demonstrate a more prominent repetition phenomenon, while Alpaca simply duplicates the provided options without providing an explanation.
> Fine-tuning on the CMExam dataset significantly reduces the repetition phenomenon and improves the overall reasonableness of the explanations. For instance, the ChatGLM-CMExam model analyzes each option in a similar manner to the solution explanation. However, some models still generate unreasonable explanations, as observed in LLaMA-CMExam, Alpaca-CMExam, and Vicuna-CMExam. This could be attributed to their training on generic data, lacking specific knowledge in the medical domain. This underscores the significance of training large language models with a focus on the medical domain."

---

> ### Author Response · Authors · 2023-08-22
> **Look forward to further feedback**
>
> Dear Reviewer Vnjt:
>
> Thanks again for all of your constructive suggestions, which have helped us improve the quality and clarity of the paper!
>
> Since the author-reviewer discussion period will end soon in 7 days, we appreciate it if you take the time to read our comments and give us some feedback. Please don't hesitate to let us know if there are any additional clarifications or experiments that we can offer, as we would love to convince you of the merits of the paper. We appreciate your suggestions. If our response resolves your concerns, we kindly ask you to consider raising the rating of our work.
>
> Thanks for your time and efforts!
>
> Best,
>
> Authors of Paper425

---

### Official Review · Reviewer_V544 · 2023-07-21
**A high-quality medical QA dataset with detailed evaluation of various LLMs.**

**Rating:** 8
**Confidence:** 4
**Clarity:** The paper exhibits a smooth and coher…

**Strengths:**

1. The proposed dataset incorporates over 60,000 Chinese medical questions, annotated across multiple dimensions using both GPT-4 and medical experts. This comprehensive annotation process provides valuable insights into the wide-ranging medical proficiencies of different models.

2. This work can serve as a benchmark for recently released large-scale models, conducting quantitative and qualitative evaluations of their performance.

3. In addition to the main analyses, supplementary experiments and assessments on few-shot learning and chain-of-thought reasoning were performed, which offer a more holistic evaluation of the models.

In conclusion, this work provides a carefully annotated medical QA dataset and a thorough evaluation of the latest models. The insights and opportunities it presents will be valuable for researchers in the fields of healthcare and NLP.

**Additional Feedback:**

Please refer to opportunities for improvement.

**Correctness:**

The CMExam dataset is constructed in a sound way, and the evaluation methods and experiment design are generally appropriate.


**Documentation:**

The authors provide explicit details regarding data collection, organization, ethical and responsible use，and a GitHub URL for reviewer access in this work. They also offer a comprehensive description of the experimental setup for the benchmark evaluation.

I have checked the dataset in the GitHub repository and found it clear and organized.

**Ethics:**

There are no ethical concerns from my perspective as the collected data does not contain user pravicy.

**Limitations:**

1. The authors have adequately addressed the limitations and potential societal impact of their work in the conclusion section.

**Opportunities For Improvement:**

1. As the parameter size of large-scale models increases, the amount of data required also grows. Therefore, there is potential for expanding the proposed dataset.
2. Since a 7B LLM (ChatGLM) can achieve a similar level as GPT-3.5 on CMExam, I am curious to know whether using LLMs with a larger parameter size can reach even higher performance and how much gap there would be compared to GPT-4.
3. It would be better to provide an English version of the dataset simuteneously, e.g., using machine translation.

**Relation To Prior Work:**

This work extensively compares the proposed CMExam dataset with previous QA datasets, highlighting the differences between them in Table 1. The authors also analyze how their benchmark evaluations of LLMs differ from other works.

**Summary And Contributions:**

This paper presents a Chinese medical QA dataset, comprised of over 60,000 instances gathered from the Chinese National Medical Licensing Examination. The dataset was meticulously annotated across five criteria, including disease groups, clinical departments, medical disciplines, areas of competency, and question difficulty levels, utilizing GPT-4 and medical experts. Furthermore, the authors evaluated the performance of 15 large language models from both the general and medical domains using this dataset, providing in-depth quantitative and qualitative analyses.

---

> ### Author Response · Authors · 2023-08-19
> **Comments by Authors**
>
> Dear Reviewer V544,
>
> We appreciate your thorough examination of our manuscript, highlighting areas of concern, and providing valuable insights to enhance our study. We have meticulously reviewed all comments and addressed each question and suggestion. We are confident that these changes have strengthened our paper. If there are any additional questions, please do not hesitate to let us know.
>
> > **Q1**: As the parameter size of large-scale models increases, the amount of data required also grows. Therefore, there is potential for expanding the proposed dataset.
>
> **A1**: We thank the reviewer for the insightful comment. We do plan to continue to expand the dataset in the future, and this includes collecting additional medical-related questions from fields like Traditional Chinese Medicine Examination and National Licensed Pharmacist Examination. We will also provide more detailed annotation to facilitate a deeper analysis of large-scale models' capabilities in the medical domain. Please keep an eye on it if you are interested :)
>
> > **Q2**: Since a 7B LLM (ChatGLM) can achieve a similar level as GPT-3.5 on CMExam, I am curious to know whether using LLMs with a larger parameter size can reach even higher performance and how much gap there would be compared to GPT-4.
>
> **A2**: Thank you for the suggestion. We are working on adding experiments for Baichuan-13B. The results will be reported and discussed once they are ready.
>
> > **Q3**: It would be better to provide an English version of the dataset simuteneously, e.g., using machine translation.
>
> **A3**: Thank you for your suggestion. In fact, there has been a study titled "Assessing ChatGPT4 Performance on Non-English National Medical Licensing Examination: A Chinese Language Evaluation" which delved into the idea of translating Chinese NMLE questions into English to enhance GPT's performance. However, the results indicated that this approach did not lead to an improvement in GPT's performance. Furthermore, while we recognize the potential benefits of having an English version, it is crucial to prioritize the authenticity and accuracy of medical content. Introducing machine translation could inadvertently introduce errors that might carry undesired consequences if not thoroughly reviewed by professionals. Additionally, the existing series of English medical licensing exams might be better suited to the needs of the English-speaking medical community.

---

> ### Author Response · Authors · 2023-08-22
> **Look forward to further feedback**
>
> Dear Reviewer V544:
>
> Thanks again for all of your constructive suggestions, which have helped us improve the quality and clarity of the paper!
>
> Since the author-reviewer discussion period will end soon in 7 days, we appreciate it if you take the time to read our comments and give us some feedback. Please don't hesitate to let us know if there are any additional clarifications or experiments that we can offer, as we would love to convince you of the merits of the paper. We appreciate your suggestions. If our response resolves your concerns, we kindly ask you to consider raising the rating of our work.
>
> Thanks for your time and efforts!
>
> Best,
>
> Authors of Paper425

---

> > ### Comment · Reviewer_V544 · 2023-08-29
> > **Thank the authors for their response.**
> >
> > I have checked the rebuttal, the revised version, and the additional feedback from other reviewers. Most of my concerns have been addressed. Notably, I find the results presented in Table 3, particularly the comparisons among GPT-3.5, GPT-4, Chat-GLM, and medical-domain LLMs, are very interesting. While it's true that more baselines can be compared as suggested by other reviewers, I think the current collection of results already provides a relatively comprehensive overview for a dataset paper.

---

### Official Review · Reviewer_989i · 2023-07-23
**While the paper presents a novel contribution to the field, several crucial elements need to be addressed to improve its clarity, comprehensiveness, and credibility.**

**Rating:** 5
**Confidence:** 4
**Clarity:** Yes, the paper is overall well writte…

**Strengths:**

The creation of CMExam presents a novel contribution to the Chinese medical LLMs community, addressing the current scarcity of evaluation resources. The paper offers valuable insights into LLM performance, highlighting the limitations and potential areas of improvement.

**Additional Feedback:**

N/A

**Correctness:**


-Data Annotation Details: More details are needed regarding the data annotation process involving medical professionals. Specifically, information about the human annotators, the annotation guidelines they followed, and the level of annotation agreement would strengthen the methodology.

-Human Performance Details: The paper lacks details on how human performance was conducted, including the number of human annotators involved and their background in domain knowledge. This information is critical for contextualizing the results.

-Missing Baselines: the Medical Domain LLM MedAlpaca should be considered as the baseline method. For the open-ended solution explanation, generative language models such as BART, and T5 should have been included in the comparison. This would provide a more comprehensive evaluation.

-Factuality Evaluation: For the open-ended solution explanation, the paper could be improved by including an evaluation of the factuality aspect of the generated responses, which is an important factor in the medical domain.

[1] Han T, Adams LC, Papaioannou JM, Grundmann P, Oberhauser T, Löser A, Truhn D, Bressem KK. MedAlpaca--An Open-Source Collection of Medical Conversational AI Models and Training Data. arXiv preprint arXiv:2304.08247. 2023 Apr 14.

**Documentation:**

Reproducibility: The paper fails to provide the code used to reproduce the results in the paper, which goes against the reproducibility guidelines provided by NeurIPS. Releasing the code would enhance the paper's credibility and allow the community to validate and build upon this work.

**Ethics:**

No.

**Limitations:**

Yes, they discussed the possible potential negative societal impact of their work.

**Opportunities For Improvement:**

While the paper presents a novel contribution to the field, several crucial elements need to be addressed to improve its clarity, comprehensiveness, and credibility. Enhancing the discussion of related work, providing more details about the methodology, and releasing the code is essential for meeting the high standards of NeurIPS. Once these issues are addressed, the paper will make a valuable addition to the community's resources.

**Relation To Prior Work:**

No.

-Comparison with Existing Resources: The paper could benefit from a more detailed comparison with existing evaluation resources, specifically the most relevant Chinese dataset MLEC-QA. Understanding the novelty and advantages of CMExam relative to these resources would strengthen the argument for its necessity.

-Related Work: The paper seems to omit many relevant studies in the related work section. References [1][2][3][4] et al should be discussed, which would provide context for the current study and ensure a comprehensive literature review.

[1] Huang Y, Bai Y, Zhu Z, Zhang J, Zhang J, Su T, Liu J, Lv C, Zhang Y, Lei J, Qi F. C-eval: A multi-level multi-discipline chinese evaluation suite for foundation models. arXiv preprint arXiv:2305.08322. 2023 May 15.

[2] Liu C, Jin R, Ren Y, Yu L, Dong T, Peng X, Zhang S, Peng J, Zhang P, Lyu Q, Su X. M3KE: A Massive Multi-Level Multi-Subject Knowledge Evaluation Benchmark for Chinese Large Language Models. arXiv preprint arXiv:2305.10263. 2023 May 17.

[3] Nori H, King N, McKinney SM, Carignan D, Horvitz E. Capabilities of gpt-4 on medical challenge problems. arXiv preprint arXiv:2303.13375. 2023 Mar 20.

[4] Fang C, Ling J, Zhou J, Wang Y, Liu X, Jiang Y, Wu Y, Chen Y, Zhu Z, Ma J, Yan Z. How does ChatGPT4 preform on Non-English National Medical Licensing Examination? An Evaluation in Chinese Language. medRxiv. 2023:2023-05.

**Summary And Contributions:**

This paper introduces CMExam, a comprehensive dataset sourced from the Chinese National Medical Licensing Examination, to evaluate large language models (LLMs) in the field of question answering (QA) in the medical domain. The dataset is intended to address gaps in evaluation resources for Chinese LLMs, and provides an innovative strategy called GPT-Assisted Annotation. The authors benchmark various LLMs, noting a considerable disparity between the performance of LLMs and human accuracy, indicating room for improvement.

---

> ### Author Response · Authors · 2023-08-19
> **Comments for Q1 and Q2**
>
> Dear Reviewer 989i,
> We appreciate your thorough examination of our manuscript, highlighting areas of concern, and providing valuable insights to enhance our study. We have meticulously reviewed all comments and addressed each question and suggestion. We are confident that these changes have strengthened our paper. If there are any additional questions, please do not hesitate to let us know.
>
> > **Q1**: More details are needed regarding the data annotation process involving medical professionals. Specifically, information about the human annotators, the annotation guidelines they followed, and the level of annotation agreement would strengthen the methodology.
>
> **A1**: We thank the reviewer for pointing out the gap in our work. Below are the details of data annotation (updated in Appendix A.6 and to our Github link at https://github.com/williamliujl/CMExam/blob/main/docs/Annotation%20Guidelines.txt)
>
> During the annotation phase, we invited one expert physician from the Second Affiliated Hospital of Zhejiang University and one senior doctoral student from Zhejiang University School of Medicine to carry out the annotations. The expert physician has over two years of clinical experience. The annotation guidelines have the following sections:
> 1. Comprehensive Question Understanding: Prior to initiating the annotation process, meticulously comprehend the medical question, ensuring a holistic grasp of its context and significance.
> 2. Subject Categorization: Identify the precise subject or medical field that the question pertains to, such as cardiology, pediatrics, or pathology.
> 3. Principal Symptoms or Medical Conditions: Ascertain and pinpoint the primary symptoms or medical conditions expounded in the question.
> 4. Examination of Pertinent Factors: Scrutinize the question for any associated factors that might be present, including the severity of the ailment, its etiology, and patient history given in the question.
> 5. Appropriate Classification System Usage: Use the accurate classification system for annotation in alignment with the determined subject and symptoms. Suitable systems could encompass the 11th revision of the International Classification of Diseases (ICD-11), the Directory of Medical Institution Diagnostic and Therapeutic Categories (DMIDTC), and others.
> 6. Addressing Multiple Annotations: In scenarios where the question encompasses multiple symptoms or medical conditions, opt for the most related classification for annotation.
> 7. Ensuring High-Quality Annotations: Adhere to the guidelines and definitions within the chosen classification system. This diligence helps avert subjectivity and ambiguity, fostering precision in the annotations.
> 8. Navigating Queries and Uncertainties: Should any doubts or uncertainties emerge during the annotation process, consult the official documents and glossaries of the chosen classification system. Engaging in discussions with professionals is also advised to achieve clarity.
> 9. Resolving Discrepancies: When disagreements emerge between annotators, a collaborative discussion shall be initiated. The objective is to reach a consensus and unify the annotation decision.
>
> > **Q2**: The paper lacks details on how human performance was conducted, including the number of human annotators involved and their background in domain knowledge. This information is critical for contextualizing the results.
>
> **A2**: We thank the reviewer for the comment. After collecting the question data from the internet, we manually matched these questions with those found in various online medical question-answering platforms to obtain the human performance on the matched questions. Therefore, the human performance we report primarily comes from the users of these platforms. Typically, these users are medical school students and others preparing for medical exams, meaning their average expertise is above that of the general population.

---

> ### Author Response · Authors · 2023-08-19
> **Comments for Q3, Q4 and Q5**
>
> > **Q3**: The Medical Domain LLM MedAlpaca should be considered as the baseline method. For the open-ended solution explanation, generative language models such as BART, and T5 should have been included in the comparison. This would provide a more comprehensive evaluation.
>
> **A3**: Thank you for the suggestion. We have incorporated additional experiments accordingly. Relevant results and discussion have been updated to the new version of our paper.
> Specifically, we fine-tuned the solution explanation task using the PromptCLUE-base(0.2B) based on the T5 architecture, and Bart-base-chinese(0.1B) + Bart-large-chinese(0.4B) based on the BART architecture. The results show that the PromptCLUE-base achieved BLEU-1 and BLEU-4 scores of 18.7 and 10.4 respectively, and ROUGE-1, ROUGE-2, and ROUGE-L scores of 40.8, 21.8, and 18.2, displaying a significant gap compared to LLMs. Both Bart-base-chinese and Bart-large-chinese outperformed the PromptCLUE-base in BLEU and ROUGE scores, but still lag behind the LLMs. This indicates that as the model parameters increase, there's a noticeable enhancement in the model's comprehension and generative capabilities. Furthermore, we evaluated the performance of MedAlpaca and its SFT on CMExam. The results showed that off-the-shelf MedAlpaca performed weakly on CMExam. However, after SFT, there was a significant improvement in performance. It achieved a ROUGE-2 score of 27.05 and a ROUGE-L score of 24.55, which are the second-best performance among all evaluation models. Detailed experiment results have been updated to the following GitHub link: https://github.com/williamliujl/CMExam/blob/main/docs/overall_comparison.jpg
>
> > **Q4**: For the open-ended solution explanation, the paper could be improved by including an evaluation of the factuality aspect of the generated responses, which is an important factor in the medical domain.
>
> **A4**: Thank you the suggestion. We have already reported the information for this part in Appendix 4. The following is the original text:
> To assess the accuracy of model-generated explanations, we conducted a study using a randomly selected sample of 50 cases in which the Language Models (LLMs) correctly predicted the answers. Medical experts were then invited to manually verify the correctness of the explanations, focusing not only on the accuracy of the answer predictions but also on the quality of the accompanying explanations. Our investigation revealed that despite the correct answer predictions by the models, certain samples exhibited errors in their corresponding explanations. These errors were categorized by the experts into three groups: explanations that were irrelevant, repeated, or inaccurate. The statistics presented in Figure 3 demonstrate that the number of samples with accurate explanations generated by the GPT models exceeded, accounting for over 90% of the total. However, it is important to note that both the ChatGLM and ChatGLM-CMExam models may produce some erroneous explanations, primarily consisting of inaccuracies and irrelevance. We have included examples of these incorrect explanations in Figure 4 in Appendix.
>
> > **Q5**: The paper could benefit from a more detailed comparison with existing evaluation resources, specifically the most relevant Chinese dataset MLEC-QA. Understanding the novelty and advantages of CMExam relative to these resources would strengthen the argument for its necessity.
>
> **A5**: We acknowledge the importance of highlighting the novel aspects of CMExam in comparison to MLEC-QA. To this end, we have created a table to demonstrate the key differences between MLEC-QA and our dataset. CMExam was designed to serve as a benchmark for LLM performance evaluation in the medical field. It distinguishes itself from MLEC-QA through annotations with authoritative references, difficulty ratings, and question explanations. Our contribution also extends beyond the dataset itself. In addition to providing CMExam, we emphasize a comprehensive benchmark of LLM performance on CMExam, a facet absent from MLEC-QA, which only compared 3 non-machine learning algorithms, 3 BERT variants, and 2 RoBERTa variants. Detailed comparison results have been updated to the following GitHub link: https://github.com/williamliujl/CMExam/blob/main/docs/Comparison_with_MLEC-QA.jpg
>
> While we don't intend to include this table directly in our manuscript at this point due to its potential narrow focus on a specific paper within our study, we have revised our related work description to encompass a more comprehensive exploration of the dissimilarities between the two datasets based on the table to underscore our distinctive contribution.

---

> ### Author Response · Authors · 2023-08-19
> **Comments for Q6 and Q7**
>
> > **Q6**: The paper seems to omit many relevant studies in the related work section. References [1][2][3][4] et al should be discussed, which would provide context for the current study and ensure a comprehensive literature review.
> > * [1] Huang Y, Bai Y, Zhu Z, Zhang J, Zhang J, Su T, Liu J, Lv C, Zhang Y, Lei J, Qi F. C-eval: A multi-level multi-discipline chinese evaluation suite for foundation models. arXiv preprint arXiv:2305.08322. 2023 May 15.
> > * [2] Liu C, Jin R, Ren Y, Yu L, Dong T, Peng X, Zhang S, Peng J, Zhang P, Lyu Q, Su X. M3KE: A Massive Multi-Level Multi-Subject Knowledge Evaluation Benchmark for Chinese Large Language Models. arXiv preprint arXiv:2305.10263. 2023 May 17.
> > * [3] Nori H, King N, McKinney SM, Carignan D, Horvitz E. Capabilities of gpt-4 on medical challenge problems. arXiv preprint arXiv:2303.13375. 2023 Mar 20.
> > * [4] Fang C, Ling J, Zhou J, Wang Y, Liu X, Jiang Y, Wu Y, Chen Y, Zhu Z, Ma J, Yan Z. How does ChatGPT4 preform on Non-English National Medical Licensing Examination? An Evaluation in Chinese Language. medRxiv. 2023:2023-05.
>
> **A6**: Thank you for sharing this valuable information. We have integrated these studies into the related work section for in-depth discussion. Here's a breakdown of our observations from these studies and a comparison of their findings with our own research:
> Both C-Eval [1] and M3KE [2] serve as multi-level multi-subject evaluation benchmarks, making them particularly suitable for assessing the capabilities of LLMs across multiple domains. In the context of the medical domain, their test datasets are comparatively small: C-Eval contains approximately 1,000 medical questions, while M3KE comprises around 3,000 medical questions, albeit lacking finer-grained annotations. Notably, both [1] and [2] focused their performance evaluations on one single model. In contrast, our study curated an extensive dataset with multi-dimension medical annotations and open-ended explanations. We also benchmarked more than 18 widely used LLMs on our dataset.
> Nori [3] undertook an evaluation of GPT-4 on various English medical datasets, such as MedQA, MedMCQA, and MultiMedQA, showcasing its state-of-the-art performance across these datasets. Furthermore, Fang [4] assessed the performance of ChatGPT on the Chinese National Medical Licensing Examination (NMLE) and concluded that ChatGPT exhibits impressive precision and reliability in handling NLME tasks in the Chinese language.
> Distinguishing itself from [3] and [4], our study involved an extensive benchmarking effort, incorporating over 18 general domain LLMs and medical domain LLMs. This comprehensive evaluation spanned both answer prediction (multiple-choice) and answer reasoning (open-ended) tasks within CMExam. Notably, our research also encompassed supervised finetuning experiments, validating that Lightweight LLMs (e.g., ChatGLM) fine-tuned using CMExam data achieved results almost on par with GPT-3.5 in answer prediction. Remarkably, these models outperformed GPT-3.5 and GPT-4 significantly in the reasoning task, all while utilizing only 3% of the parameters found in GPT-3.5.
>
> > **Q7**: The paper fails to provide the code used to reproduce the results in the paper, which goes against the reproducibility guidelines provided by NeurIPS. Releasing the code would enhance the paper's credibility and allow the community to validate and build upon this work.
>
> **A7**: Sorry for our oversight. We have uploaded all code and detailed guidelines to the following GitHub link:
> https://github.com/williamliujl/CMExam

---

> ### Author Response · Authors · 2023-08-22
> **Look forward to further feedback**
>
> Dear Reviewer 989i:
>
> Thanks again for all of your constructive suggestions, which have helped us improve the quality and clarity of the paper!
>
> Since the author-reviewer discussion period will end soon in 7 days, we appreciate it if you take the time to read our comments and give us some feedback. Please don't hesitate to let us know if there are any additional clarifications or experiments that we can offer, as we would love to convince you of the merits of the paper. We appreciate your suggestions. If our response resolves your concerns, we kindly ask you to consider raising the rating of our work.
>
> Thanks for your time and efforts!
>
> Best,
>
> Authors of Paper425

---

### Author Response · Authors · 2023-08-20
**Summary of Changes and Revised Manuscript**

We are grateful for all the reviewers' detailed reviews and constructive comments. In response to the reviewers' feedback, we have updated the manuscript where changes are colored blue.

In the latest version, we have:

1. Integrated several related studies into the paper for in-depth discussion. (See Section 1)
2. Added a detailed comparison of the CMExam and MLEC-QA datasets, including Annotation content, References, and Number of unique values. (See Section 2).
3. For the answer prediction and solution explanation tasks, we incorporated:
   - PromptCLUE-base(0.2B) based on the T5 architecture
   - Bart-base-chinese(0.1B) + Bart-large-chinese(0.4B) based on the BART architecture.
   - MedAlpaca(7B)

    Additionally, we've provided their results after sft on the CMExam dataset. (See Table 3)
4. Presented details of the data annotation guideline. (See Appendix A.6)
5. Expounded upon our explorations and results regarding prompt design in the pre-annotation phase. (See Appendix A.7)

We believe that this updated manuscript has greatly improved by addressing the reviewers' concerns.

---

### Decision · Program_Chairs · 2023-09-22

**Decision:**

Accept (Poster)

**Comment:**

Reviewers are generally positive in recommending the acceptance of this manuscript but also raise concerns such as the comprehensiveness of the benchmark and the use of larger LLMs.